# Skyscrapers in the Twenty-First Century City: A Global Snapshot

**Kheir Al-Kodmany**

Department of Urban Planning and Policy, College of Urban Planning and Public Affairs, University of Illinois, Chicago, IL 60607, USA; kheir@uic.edu

**Abstract:** The first two decades of the twenty-first century represent a major milestone in skyscraper developments. By analyzing extensive data, the research presented here contrasts building activities of skyscrapers before and after the turn of the 21st century. It examines tall buildings in the world's major continents (Asia, Europe, North America, Oceana, Middle East, South America, Central America, and Africa) and their respective cities including Shanghai, Beijing, Shenzhen, Bangkok, London, Moscow, New York, Chicago, Miami, San Francisco, Melbourne, Sydney, Dubai, Doha, Riyadh, Tel Aviv, São Paulo, Panama City, Mexico City, and Nairobi. By using nearly 40 tables and 80 maps, the paper highlights the rapid activities of building significant skyscrapers at greater heights, elucidates the changes in functions and services, and delineates shifts in spatial patterns and visual impact.

**Keywords:** emerging and developed skyscraper cities; increased height; greater counts; new morphology

## 1. Introduction

This paper reviews recent skyscraper developments globally. In the past two decades, the world has witnessed an unprecedented construction boom of tall buildings. Despite financial difficulties and construction challenges, cities have been building an increasingly greater number of tall buildings with greater heights. This paper intends to map out recent construction developments in global cities and attempts to answer basic questions, which are shown below.

- What are the most active regions in constructing tall buildings?
- What are the tallest buildings?
- What are the driving forces of building skyscrapers?
- Are there changes in the functionality of this building typology?
- How do new skyscraper developments alter the spatial patterns of their respective cities?

For answering these questions, the paper conducts an extensive literature review and uses the Skyscraper Center's database of the Council on Tall Buildings and Urban Habitat (CTBUH) [1]. Therefore, all data reported in this paper including the building's height, use, number of floors, square footages, and date of project completion are based on this database. Reported building heights refer to the architectural tip of the building and the reported numbers of floors refer to those that are above ground. All tables and maps in this paper are produced by the same database. Each city is represented by four maps grouped with the following order. Two maps at the top compare the larger geographic coverage and two maps at the bottom compare the smaller geographic coverage of the same city between 2000 and 2020. These comparisons highlight changes and recent activities regarding building skyscrapers in these cities.

Detailed definitions of tall buildings are offered in previous research [2,3]. For the sake of brevity, this paper embraces the following definitions.

1.　A tall building, a high-rise, or a tower is a 50 m+ building.
2.　A skyscraper is a 100 m+ building.
3.　A supertall or ultra-tall is a 300 m+ building.
4.　A megatall is a 600 m+ building.

Regarding building status, the paper embraces the CTBUH's definitions as follows [3].

1.　Under construction. Once construction workers complete clearing the site and finish the foundation, the building earns the "under construction" title.
2.　Structurally topped out. Once the highest primary structural element of a building under construction is in place, the building earns the "structurally topped out" title.
3.　Architecturally topped out. When a building under construction reaches its full height both structurally and architecturally (e.g., including its spire and parapets), it earns the "architecturally topped out" title.
4.　On hold. When construction work is suspended indefinitely, the building will be called "on hold" if the intention to complete construction to its original design is retained.
5.　Never completed. When construction work is suspended and there is an intention to complete the project with different plans than original ones, the research calls this building "never completed."
6.　Proposed. When a building fulfills the following criteria, it earns the "proposed" title:

- The project has a specific site, an owner, and a developer who are seriously interested in executing the project.
- The professional design and planning team has passed the conceptual design stage of the project and are progressing toward completing construction drawings.
- The project has obtained construction permission or in the process to do so.
- The announcement of the "proposed" building comes from a credible source.

7.　*Vision*. A building project earns this title when it meets one of the following three criteria:

- A project idea that does not fulfill the "proposed" criteria mentioned above.
- A "proposed" project that developers could not advance to the construction stage, or
- A project idea whom architects conceived to be an inspirational proposition.

8.　*Demolished*. When a building is destroyed by an authority at the end of its life cycle or by a nature-made (e.g., earthquake, hurricane, etc.) or a human-made disaster (e.g., terrorist attack, war, etc.), it earns the "demolished" title.

The paper holds the following structure. First, it provides a global overview of the rapid pace of constructing tall buildings, highlighting major activities, and the tallest skyscrapers. Then it discusses the driving forces of building tall construction projects. Next, it examines case studies within each world continent. The examined case studies compare the spatial patterns and tallest skyscrapers in 2000 and 2020. Afterward, the paper discusses key findings and closes with a summary conclusion.

*Overview of Constructing Tall Buildings*

Simply put, in the first two decades of the 21st century, cities will add 8827 tall buildings to the 7513 they built previously. Construction workers are currently erecting 2816 tall buildings while developers have proposed 3043 tall building projects. Furthermore, architects and planners have presented 2112 visionary projects [1–3]. Regarding height, cities built merely 24 supertalls (300 m+ buildings) before 2000. In contrast, they have built 144 supertalls since then. This count increases significantly when categories other than the completed ones are included. Cities also have built three megatalls (600 m+ buildings) in recent years. However, they constructed none before. CTBUH research notes, "In the 70 years before 9/11, the record for the tallest building grew 230 feet. Since then, it has shot up 1234 feet... It is poised to rise much higher over the next decade" [4]. Buildings under

construction will reach greater heights and both proposed and visionary projects suggest building even higher. Projecting these planning and construction activities into the future reveals that cities will host thousands of taller buildings (Table 1) [1].

**Table 1.** Tall buildings in 2000 and 2020. In recent years, we have been building more tall buildings with greater heights. In addition, buildings "under construction," "proposed," or "visionary" categories feature greater heights. Shaded rows highlight the supertall and megatall categories.

|  | 2000 | 2000-2020 | | | |
| --- | --- | --- | --- | --- | --- |
| Building Height (m) | Completed | Completed | Under Construction | Proposed | Visionary |
| 50+ | 7513 | 8827 | 2816 | 3043 | 2112 |
| 100+ | 3649 | 5117 | 1623 | 1266 | 1730 |
| 150+ | 1139 | 2715 | 1111 | 809 | 1643 |
| 200+ | 261 | 924 | 627 | 469 | 1108 |
| 250+ | 68 | 255 | 302 | 285 | 749 |
| 300+ | 24 | 144 | 150 | 174 | 527 |
| 350+ | 10 | 29 | 58 | 87 | 361 |
| 400+ | 4 | 16 | 32 | 51 | 259 |
| 450+ | 2 | 9 | 20 | 27 | 195 |
| 500+ | 0 | 6 | 13 | 19 | 156 |
| 550+ | 0 | 3 | 6 | 14 | 114 |
| 600+ | 0 | 3 | 2 | 8 | 93 |

Particularly, the last few years have witnessed an accelerated pace of constructing tall buildings. According to the *CTBUH's 2015 Year in Review Report* [4]: "It's clear that 2015 was a banner year for skyscrapers: Across the globe in that year alone, 106 tall buildings (above 200 meters or 656 feet in height) were finished, surpassing 2014's previous record of 99." The year 2016 witnessed even greater activities in constructing tall buildings than previous years. According to the *CTBUH's 2016 Year in Review Report* [5]: "128 buildings of 200 meters' height or greater were completed around the world in 2016—setting a new record for annual tall building completions and marking the third consecutive record-breaking year. The 128 buildings completed in 2016 beat every previous year on record including the previous record high of 114 completions in 2015." Similarly, the year 2017 has witnessed constructing even greater numbers of tall buildings of all categories. For example, in the 200 m+ category, the world built 144 buildings, which is higher by 128 buildings in 2016 and, in the 300 m+ category, it built 126 buildings, which is up from 111 in 2016 [6].

Certainly, taller buildings attract global attention. The tallest building in the world until 1998 was the 442 m (1451 ft) high Willis Tower (formerly Sears Tower) in Chicago, Illinois. The Petronas Towers in Kuala Lumpur, Malaysia has taken the title in 1998 by standing at 452 m (1483 ft). Soon after that, in 2004, Taipei 101 in Taipei, Taiwan surpassed the Petronas Towers by soaring to a height of 509 m (1670 ft) to become the world's tallest building. It retained the title until Dubai completed Burj Khalifa in 2010, which rises to a height of 828 m (2717 ft) [2].

Completed in 2015, the Shanghai Tower in Shanghai, China rises to 632 m (2074 ft) and is the world's second tallest building. Recently, Seoul has completed the 555 m (1821 ft) Lotte World Tower, which became the tallest building in South Korea and the world's fifth tallest. Nevertheless, Jeddah Tower (formerly Kingdom Tower) in Jeddah, Saudi Arabia will surpass Burj Khalifa by reaching an unprecedented height of 1000 m (3280 ft) and will become the world's tallest building when completed in 2020 [2].

Architects and engineers believe that we can build even taller than the Jeddah Tower. For example, William Baker, a chief structural engineer for Skidmore, Owings, and Merrill (SOM), explains in a recent article titled "Is there a limit to how tall buildings can get?" that "We could easily do a mile . . . We could do probably quite a bit more" [7]. It is remarkable to consider that it took 80 years (1930–2010) to build the first 50 supertalls while it took merely five years (2010–2015) to build the next

50 supertalls [8] (p. 38). Some of the tall buildings under construction will soar even higher than the existing tallest buildings [9] (Table 2).

**Table 2.** The world's 10 tallest skyscrapers in 2020. Note that all these buildings will be in Asia and the Middle East—North America will have none.

| # | Building Name | City | Height (m) | Floors | Completion |
|---|---------------|------|-----------|--------|-----------|
| 1 | Jeddah Tower | Jeddah (SA) | 1000 | 167 | 2020 |
| 2 | Burj Khalifa | Dubai (AE) | 828 | 163 | 2010 |
| 3 | Wuhan Greenland Center | Wuhan (CN) | 636 | 125 | 2018 |
| 4 | Shanghai Tower | Shanghai (CN) | 632 | 128 | 2015 |
| 5 | Merdeka PNB118 | Kuala Lumpur (MY) | 630 | 118 | 2020 |
| 6 | Makkah Royal C lock Tower | Mecca (SA) | 601 | 120 | 2012 |
| 7 | Ping An Finance Center | Shenzhen (CN) | 599 | 115 | 2017 |
| 8 | Goldin Finance 117 | Tianjin (CN) | 597 | 128 | 2018 |
| 9 | Global Financial C enter Tower 1 | Shenyang (CN) | 568 | 114 | 2018 |
| 10 | Lotte World Tower | Seoul (KR) | 555 | 123 | 2017 |

Continental shares reveal that North America has championed constructing tall buildings in the 20th century by constructing about 3867 buildings while other continents lagged (see Table 3).

- North America: 3867
- Asia: 1593
- Europe: 968
- Oceania: 473
- South America: 286
- Africa: 132
- Middle East: 106
- Central America: 88

However, in the 21st century, tall building developments exhibit a geographic shift, as follows.

- Asia: 3962
- North America: 1710
- Europe: 1426
- Middle East: 647
- Oceania: 473
- South America: 350
- Central America: 201
- Africa: 50

**Table 3.** Tall buildings in 2000 and 2020. The table organizes data based on the world's continents, which is listed alphabetically. Note the geographic shift from North America to Asia.

| Continent | # of Tall Buildings Completed in 2000 | # of Tall Buildings Completed in 2020 | Under Construction | Proposed | Visionary |
|-----------|---------------------------------------|---------------------------------------|--------------------|----------|-----------|
| Africa | 132 | 50 | 24 | 19 | 30 |
| Asia | 1593 | 3962 | 1275 | 845 | 663 |
| Central America | 88 | 201 | 27 | 20 | 34 |
| Europe | 968 | 1426 | 597 | 801 | 510 |
| Middle East | 106 | 647 | 220 | 131 | 200 |
| North America | 3867 | 1710 | 404 | 719 | 510 |
| Oceania | 473 | 465 | 127 | 500 | 109 |
| South America | 286 | 350 | 53 | 29 | 56 |

Overall, Asia accounts for three-quarters of the newly completed tall buildings of the world. In addition, "under construction," "proposed," and "visionary" categories demonstrate that Asia takes the lead. Regarding height, Asia has also been championing the race (Table 4). It has built more supertalls (300 m+) in recent years than any other continents.

- Asia: 48
- Middle East: 27
- North America: 6
- Europe: 5
- Oceania: 1
- South America: 0
- Central America: 0
- Africa: 0

**Table 4.** Tall buildings in 2000 and 2020. The breakdown is based on world continents and buildings' heights.

| Continent | Height (m) | # of Tall Buildings Completed in 2000 | # of Tall Buildings Completed in 2020 | Under Construction | Proposed | Visionary |
|---|---|---|---|---|---|---|
| Africa | 50+ | 132 | 50 | 24 | 19 | 30 |
| | 100+ | 83 | 30 | 13 | 6 | 25 |
| | 150+ | 4 | 3 | 5 | 5 | 23 |
| | 200+ | 1 | 0 | 1 | 5 | 17 |
| | 250+ | 0 | 0 | 0 | 4 | 13 |
| | 300+ | 0 | 0 | 0 | 3 | 10 |
| | 350+ | 0 | 0 | 0 | 2 | 8 |
| | 400+ | 0 | 0 | 0 | 1 | 7 |
| | 450+ | 0 | 0 | 0 | 1 | 5 |
| | 500+ | 0 | 0 | 0 | 1 | 2 |
| | 550+ | 0 | 0 | 0 | 0 | 2 |
| | 600+ | 0 | 0 | 0 | 0 | 1 |
| Asia | 50+ | 1593 | 3788 | 1275 | 845 | 663 |
| | 100+ | 940 | 2657 | 881 | 407 | 620 |
| | 150+ | 488 | 1875 | 747 | 337 | 599 |
| | 200+ | 95 | 645 | 463 | 234 | 461 |
| | 250+ | 24 | 162 | 228 | 164 | 339 |
| | 300+ | 10 | 48 | 117 | 111 | 262 |
| | 350+ | 7 | 12 | 45 | 63 | 188 |
| | 400+ | 3 | 9 | 24 | 40 | 134 |
| | 450+ | 2 | 6 | 16 | 22 | 98 |
| | 500+ | 0 | 3 | 11 | 16 | 80 |
| | 550+ | 0 | 1 | 6 | 11 | 51 |
| | 600+ | 0 | 1 | 2 | 7 | 41 |
| Central America | 50+ | 88 | 201 | 27 | 20 | 34 |
| | 100+ | 36 | 136 | 20 | 13 | 31 |
| | 150+ | 10 | 68 | 13 | 9 | 30 |
| | 200+ | 1 | 26 | 6 | 4 | 19 |
| | 250+ | 0 | 4 | 3 | 4 | 14 |
| | 300+ | 0 | 0 | 0 | 3 | 8 |
| | 350+ | 0 | 0 | 0 | 0 | 3 |
| | 400+ | 0 | 0 | 0 | 0 | 1 |
| | 450+ | 0 | 0 | 0 | 0 | 0 |
| | 500+ | 0 | 0 | 0 | 0 | 0 |
| | 550+ | 0 | 0 | 0 | 0 | 0 |
| | 600+ | 0 | 0 | 0 | 0 | 0 |

**Table 4.** *Cont.*

| Continent | Height (m) | # of Tall Buildings Completed in 2000 | # of Tall Buildings Completed in 2020 | Under Construction | Proposed | Visionary |
|---|---|---|---|---|---|---|
| Europe | 50+ | 968 | 1426 | 597 | 801 | 510 |
| | 100+ | 316 | 527 | 161 | 194 | 320 |
| | 150+ | 42 | 132 | 63 | 71 | 260 |
| | 200+ | 11 | 34 | 19 | 22 | 154 |
| | 250+ | 2 | 9 | 10 | 6 | 98 |
| | 300+ | 0 | 5 | 4 | 4 | 55 |
| | 350+ | 0 | 1 | 2 | 1 | 31 |
| | 400+ | 0 | 0 | 2 | 1 | 18 |
| | 450+ | 0 | 0 | 1 | 0 | 10 |
| | 500+ | 0 | 0 | 0 | 0 | 8 |
| | 550+ | 0 | 0 | 0 | 0 | 7 |
| | 600+ | 0 | 0 | 0 | 0 | 6 |
| Middle East | 50+ | 106 | 647 | 220 | 131 | 200 |
| | 100+ | 47 | 426 | 117 | 50 | 159 |
| | 150+ | 11 | 283 | 85 | 33 | 152 |
| | 200+ | 4 | 135 | 65 | 27 | 105 |
| | 250+ | 4 | 58 | 36 | 19 | 70 |
| | 300+ | 3 | 27 | 18 | 16 | 59 |
| | 350+ | 1 | 12 | 9 | 11 | 47 |
| | 400+ | 0 | 4 | 5 | 3 | 34 |
| | 450+ | 0 | 2 | 2 | 3 | 29 |
| | 500+ | 0 | 2 | 2 | 2 | 22 |
| | 550+ | 0 | 2 | 1 | 2 | 21 |
| | 600+ | 0 | 2 | 1 | 1 | 16 |
| North America | 50+ | 3867 | 1710 | 404 | 719 | 510 |
| | 100+ | 1959 | 995 | 273 | 338 | 455 |
| | 150+ | 522 | 267 | 134 | 214 | 436 |
| | 200+ | 134 | 63 | 58 | 105 | 252 |
| | 250+ | 35 | 18 | 22 | 61 | 157 |
| | 300+ | 11 | 6 | 12 | 30 | 96 |
| | 350+ | 2 | 4 | 4 | 7 | 58 |
| | 400+ | 1 | 3 | 2 | 4 | 46 |
| | 450+ | 0 | 1 | 1 | 0 | 41 |
| | 500+ | 0 | 1 | 0 | 0 | 34 |
| | 550+ | 0 | 0 | 0 | 0 | 25 |
| | 600+ | 0 | 0 | 0 | 0 | 24 |
| Oceania | 50+ | 473 | 465 | 127 | 500 | 109 |
| | 100+ | 186 | 174 | 67 | 240 | 101 |
| | 150+ | 43 | 54 | 26 | 123 | 90 |
| | 200+ | 13 | 19 | 11 | 62 | 54 |
| | 250+ | 3 | 3 | 6 | 21 | 32 |
| | 300+ | 0 | 1 | 1 | 3 | 17 |
| | 350+ | 0 | 0 | 0 | 0 | 10 |
| | 400+ | 0 | 0 | 0 | 0 | 7 |
| | 450+ | 0 | 0 | 0 | 0 | 4 |
| | 500+ | 0 | 0 | 0 | 0 | 3 |
| | 550+ | 0 | 0 | 0 | 0 | 2 |
| | 600+ | 0 | 0 | 0 | 0 | 1 |
| South America | 50+ | 286 | 350 | 53 | 29 | 56 |
| | 100+ | 137 | 182 | 39 | 21 | 54 |
| | 150+ | 19 | 33 | 24 | 16 | 53 |
| | 200+ | 2 | 2 | 12 | 9 | 46 |
| | 250+ | 0 | 1 | 4 | 4 | 26 |
| | 300+ | 0 | 1 | 0 | 1 | 20 |
| | 350+ | 0 | 0 | 0 | 1 | 16 |
| | 400+ | 0 | 0 | 0 | 1 | 12 |
| | 450+ | 0 | 0 | 0 | 1 | 8 |
| | 500+ | 0 | 0 | 0 | 0 | 7 |
| | 550+ | 0 | 0 | 0 | 0 | 6 |
| | 600+ | 0 | 0 | 0 | 0 | 6 |

The "under construction" category reveals that Asia surpasses all continents by a large margin.

- Asia: 117
- Middle East: 18
- North America: 12
- Europe: 4
- Oceania: 1
- Central America: 1
- South America: 0
- Africa: 0

The "proposed" category reveals a similar order, as shown below.

- Asia: 111
- North America: 30
- Middle East: 16
- Europe: 4
- Central America: 3
- Oceania: 3
- Africa: 3
- South America: 1

The "megatall" category shows that the Middle East has constructed two megatalls (Burj Khalifa and Makkah Royal Clock Tower) and Asia built one (Shanghai Tower). Asia is now constructing three megatalls including Wuhan CTF Finance Center in Wuhan, China, Merdeka PNB118 in Kuala Lumpur, Maylasia, and Grand Rama 9 Tower in Bangkok, Taiwan while the Middle East is constructing Jeddah Tower in Jeddah, Saudi Arabia. Other continents are constructing none. As expected, visionary projects "dream" of constructing taller buildings in every continent.

Among all Asian countries, China stands out as the epicenter of high-rise construction. Many of the towers that China built or are under construction have lent a global recognition. Over the past two decades, tall building construction has increased dramatically in China, which is a phenomenon that moved from first-tier to second-tier and third-tier cities. In this regard, Quanhong Li (2017) explains, "High-density vertical urban developments are shaping the future identities of these cities by way of their strategic locations, massive scale, significant functional mix, and large social, economic, and environmental impact" [9] (p. 32). Meanwhile, tall building projects have been proliferating in established skyscraper cities.

Starting during the 21st century, China has built 52% of all Asia's tall buildings (1995/3788). The rest of Asia (31 countries) shares the remaining 48%. The CTBUH enlightens that China leads the world by concentrating on completions of 200 m+ (656 ft+) buildings. While in 2015, China counted for 58% of the world's tall buildings of this height category (62/106), in 2016, it counted for 66% (84/128) [5]. Specifically, China achieved significant leaps in this height category in the past few years where the total number of 200 m+ (656 ft+) buildings more than doubled between 2000 and 2020. Remarkably, despite expecting an economic slowdown for China, the CTBUH predicts that China will continue to rank first in building skyscrapers for years to come. "With the closest national contender, the US, in 2016 having only seven completions and China having 84, it is clear that the gap will take a number of years to close, if it ever does" [10].

European cities that have historically banned tall buildings to protect their valuable built heritage—e.g., London, Paris, Frankfurt, Amsterdam, Moscow, and Warsaw—have resumed constructing tall buildings. In her article "Development of High-Rise Buildings in Europe in the 20th and 21st centuries," Joanna Pietrzak writes [11] (p. 31), "It was not until the 1950s that Europe began to construct buildings taller than 100 meters. By 2013, skyscrapers had been constructed in over

100 European cities located in 30 different countries, and the trend towards the expansion of high-rise construction continues." European cities have completed a record number of skyscrapers since the turn of the 21st century and this trend is likely to continue in the coming years. Europe had merely two 250 m+ (820 ft+) tall buildings before the 21st century. However, by 2020, Europe will have 23 buildings of this height category.

Since the turn of the 21st century, Russian cities have been experiencing a period of extensive high-rise construction. Russian cities such as Moscow, St. Petersburg, and Groznyj have been constructing some of Europe's tallest buildings. This construction boom happened partially because of the prosperous recovery of the national economy after the 1998 crisis. These cities are trying to convey to the world that the country is reclaiming its glory and joining the global economy. After the 20th century, Russia has built fourteen 200 m+ (656 ft+) buildings, which equals 40% of all Europe's new buildings of this height category (14/35). There are now six buildings of this category under construction as well. Currently, Russia has seven of the 10 tallest skyscrapers in Europe and will have eight of the 10 tallest buildings in 2020. Overall, 64% of the tallest buildings in Russia were built in the 2000s [2].

Sustainable land use and transportation are different from the suburban auto-dependent sprawl development that has dominated North America's landscape since 1950. Compact, mixed use, walkable, transit-oriented places offer significant environmental, economic, and social benefits. Consequently, major cities in North America such as New York City, Chicago, Miami, Philadelphia, Seattle, San Francisco, Los Angeles, Toronto, Vancouver, and Calgary have also been constructing significant tall buildings around mass transit nodes. Australian cities are also progressively building upward due to multiple reasons such as increased land value. Sydney, Melbourne, Gold Coast, and Brisbane are taking the lead.

Significantly, before the turn of the century, the Middle East built only 106 tall buildings and in the first two decades of the 21st century, it will construct over 647 tall buildings. Currently, the Middle East has 28 supertalls and two megatalls. By 2020, it will have 46 supertalls and three megatalls. Cities in the Middle East such as Dubai, Abu Dubai, Doha, Jeddah, Mecca, Riyadh, Kuwait, Tel Aviv, and Beirut have been among the most active in building significant skyscrapers. For example, Mecca and Dubai have built two of the world's three megatalls, Makkah Royal Clock Tower and Burj Khalifa, respectively—the third is Shanghai Tower in Shanghai, China. By 2020, Jeddah will add to the Middle East another megatall, which is the new world's tallest building called Jeddah Tower soaring 1000 m (3280 ft). Doha, Qatar had no skyscrapers before the turn of the century and, by 2020, it will have completed 39 skyscrapers. Likewise, Abu Dhabi, UAE, had merely one skyscraper before the turn of the century and now has 31 skyscrapers. Similarly, before the turn of the century, Manama, Bahrain had no skyscrapers. However, it has now 15 skyscrapers.

Since the beginning of the 21st Century, South America has built 350 tall buildings—more than it did before (286 buildings). The most active cities in building skyscrapers are Santiago (Chile), Bucaramanga, Cartagena (Colombia), Buenos Aires (Argentina), São Paulo, Balneario Camboriu, and Curitiba (Brazil). Notably, in 2014, South America built its two tallest towers including the 300 m (1000 ft) Torre Costanera in Santiago (Chile) and the 219 m (719 ft) Millennium Palace in Balneario Camboriu (Brazil). However, there are six buildings currently under construction of remarkable heights including: One Tower (280 m (919 ft)), Yachthouse Residence Club towers (270 m each (886 ft)), Infinity Coast Tower (237 m (778 ft)), Epic Tower (220 m (722 ft)), and Boreal Tower (220 m (772 ft)) [2,11]. Although South America will continue to build taller, it lags significantly on buildings' height when compared to other continents including Asia, Middle East, North America, and Europe.

Central America has also been active in constructing tall buildings. Since the turn of the century, it built 201 tall buildings, which is more than it did before (88 buildings). Regarding height, Central America built all of its 26 200 m+ buildings post the turn of the century except one. Among the cities constructing tall buildings are Panama City (Panama) and Mexico City, Monterrey, and Guadalajara

(Mexico) [2]. These cities surpass other cities in the continent regarding counts and heights. Currently, the continent has no supertall and will have no supertall by 2020 as well [12,13].

Comparing to other continents, Africa has the smallest share of tall buildings. It has constructed 182 tall buildings, 132 of which were completed before the turn of the century and 50 after. Regarding height, its buildings are shorter than that of other continents. It has no supertall and merely one building that exceeds 200 m (656 ft). By 2020, Africa will also have no supertall and will have merely two 200 m+ buildings. Cities engaged in constructing tall buildings include Johannesburg, Pretoria, and Sandton (South Africa), Dar es Salaam (Tanzania), Nairobi (Kenya), and Lagos (Nigeria) [14,15]. Overall, the pace of construction is slow and fewer buildings are of significant height in these cities.

In short, the entire world is witnessing an upsurge of building skyscrapers with increasing heights. Tables 5–12 highlight the tallest skyscrapers in the world's continents in 2020 and Figures 1 and 2 illustrate some of these examples.

**Table 5.** The Middle East's 10 tallest skyscrapers in 2020.

| # | Building Name | City | Height (m) | Floors | Completion |
|---|---|---|---|---|---|
| 1 | Jeddah Tower | Jeddah (SA) | 1000 | 167 | 2020 |
| 2 | Burj Khalifa | Dubai (AE) | 828 | 163 | 2010 |
| 3 | Makkah Royal Clock Tower | Mecca (SA) | 601 | 120 | 2012 |
| 4 | Entisar Tower | Dubai (AE) | 520 | 111 | 2020 |
| 5 | Marina 106 | Dubai (AE) | 445 | 104 | 2019 |
| 6 | Diamond Tower | Jeddah (SA) | 432 | 93 | 2019 |
| 7 | Marina 101 | Dubai (AE) | 427 | 101 | 2017 |
| 8 | Princess Tower | Dubai (AE) | 413 | 101 | 2012 |
| 9 | Al Hamra Tower | Kuwait City (KW) | 413 | 80 | 2011 |
| 10 | 23 Marina | Dubai (AE) | 392 | 88 | 2012 |

**Table 6.** Asia's 10 tallest skyscrapers in 2020.

| # | Building Name | City | Height (m) | Floors | Completion |
|---|---|---|---|---|---|
| 1 | Wuhan Greenland Center | Wuhan | 636 | 125 | 2020 |
| 2 | Shanghai Tower | Shanghai | 632 | 128 | 2015 |
| 3 | Ping An Finance Center | Shenzhen | 599 | 115 | 2017 |
| 4 | Goldin Finance 117 | Tianjin | 597 | 128 | 2018 |
| 5 | Global Financial Center Tower 1 | Shenyang | 568 | 114 | 2018 |
| 6 | Guangzhou CTF Finance Center | Guangzhou | 530 | 111 | 2016 |
| 7 | Tianjin CTF Finance Center | Tianjin | 530 | 97 | 2018 |
| 8 | China Zun Tower | Beijing | 528 | 108 | 2018 |
| 9 | Dalian Greenland Center | Dalian | 518 | 88 | 2019 |
| 10 | International Commerce Center | Hong Kong | 484 | 108 | 2010 |

**Table 7.** North America's 10 tallest skyscrapers in 2020.

| # | Building Name | City | Height (m) | Floors | Completion |
|---|---|---|---|---|---|
| 1 | One World Trade Center | New York City (US) | 541 | 94 | 2014 |
| 2 | Central Park Tower | New York City (US) | 472 | 95 | 2019 |
| 3 | 111 West 57th Street | New York City (US) | 438 | 80 | 2018 |
| 4 | 432 Park Avenue | New York City (US) | 426 | 85 | 2015 |
| 5 | 30 Hudson Yards | New York City (US) | 387 | 73 | 2019 |
| 6 | Vista Tower | Chicago (US) | 362 | 98 | 2020 |
| 7 | Comcast Technology Center | Philadelphia (US) | 342 | 59 | 2018 |
| 8 | Wilshire Grand Center | Los Angeles (US) | 335 | 73 | 2017 |
| 9 | 3 World Trade Center | New York City (US) | 329 | 69 | 2018 |
| 10 | Salesforce Tower | San Francisco (US) | 326 | 61 | 2018 |

**Table 8.** Europe's 10 tallest skyscrapers in 2020.

| # | Building Name | City | Height (m) | Floors | Completion |
|---|---|---|---|---|---|
| 1 | Lakhta Center | St. Petersburg (RU) | 462 | 86 | 2018 |
| 2 | Multifunctional High-Rise Complex—Akhmat Tower | Groznyj (RU) | 435 | 102 | 2020 |
| 3 | Federation Towers—Vostok Tower | Moscow (RU) | 374 | 95 | 2016 |
| 4 | OKO—Residential Tower | Moscow (RU) | 354 | 90 | 2015 |
| 5 | Mercury City Tower | Moscow (RU) | 339 | 75 | 2013 |
| 6 | NEVA TOWERS 2 | Moscow (RU) | 338 | 77 | 2020 |
| 7 | Varso Tower | Warsaw (PL) | 310 | 53 | 2020 |
| 8 | Stalnaya Vershina | Moscow (RU) | 309 | 72 | 2015 |
| 9 | The Shard | London (GB) | 306 | 73 | 2013 |
| 10 | Capital City Moscow Tower | Moscow (RU) | 302 | 76 | 2010 |

**Table 9.** Oceana's 10 tallest skyscrapers in 2020.

| # | Building Name | City | Height (m) | Floors | Completion |
|---|---|---|---|---|---|
| 1 | Australia 108 | Melbourne (AU) | 317 | 100 | 2020 |
| 2 | Crown Sydney Hotel and Resort | Sydney (AU) | 271 | 75 | 2019 |
| 3 | Brisbane Sky Tower | Brisbane (AU) | 270 | 89 | 2018 |
| 4 | Aurora Melbourne Central | Melbourne (AU) | 269 | 88 | 2019 |
| 5 | 300 George Street Tower 1 | Brisbane (AU) | 262 | 81 | 2018 |
| 6 | 1 William Street | Brisbane (AU) | 260 | 46 | 2017 |
| 7 | Prima Pearl Apartments | Melbourne (AU) | 254 | 72 | 2014 |
| 8 | Infinity | Brisbane (AU) | 249 | 81 | 2014 |
| 9 | Victoria One | Melbourne (AU) | 246 | 75 | 2018 |
| 10 | Soleil | Brisbane (AU) | 243 | 79 | 2012 |

**Table 10.** South America's 10 tallest skyscrapers in 2020.

| # | Building Name | City | Height | Floors | Completion |
|---|---|---|---|---|---|
| 1 | Torre Costanera | Santiago (CL) | 300 | 62 | 2014 |
| 2 | Yacht House Residence Club by Pininfarina Tower 1 | Balneario Camboriu (BR) | 270 | 75 | 2018 |
| 3 | Yachthouse Residence Club by Pininfarina Tower 2 | Balneario Camboriu (BR) | 270 | 75 | 2018 |
| 4 | One Tower | Balneario Camboriu (BR) | 263 | 77 | 2022 |
| 5 | BD Bacata Torre 1 | Bogota (CO) | 260 | 67 | 2017 |
| 6 | La Isla Multiespacio Torre Officinas | Valencia (VE) | 244 | 55 | 2017 |
| 7 | Infinity Coast Tower | Balneario Camboriu (BR) | 237 | 66 | 2017 |
| 8 | Alvear Tower Puerto Madero | Buenos Aires (AR) | 235 | 54 | 2017 |
| 9 | Parque Central Torre Officinas I | Caracas (VE) | 225 | 56 | 1979 |
| 10 | Parque Central Torre Officinas II | Caracas (VE) | 225 | 56 | 1983 |

**Table 11.** Central America's 10 tallest skyscrapers in 2020.

| # | Building Name | City | Height | Floors | Completion |
|---|---|---|---|---|---|
| 1 | Trump Ocean Club International Hotel & Tower | Panama City (PA) | 284 | 70 | 2011 |
| 2 | Torre Vitri | Panama City (PA) | 281 | 75 | 2012 |
| 3 | Torre Koi | Monterrey (MX) | 279 | 67 | 2017 |
| 4 | Torre Mitikah | Mexico City (MX) | 267 | 62 | - |
| 5 | Bicsa Financial Center | Panama City (PA) | 267 | 66 | 2013 |
| 6 | The Point | Panama City (PA) | 266 | 67 | 2011 |
| 7 | YooPanama Inspired by Starck | Panama City (PA) | 247 | 78 | 2013 |
| 8 | Torre Reforma | Mexico City (MX) | 246 | 56 | 2016 |
| 9 | Ocean Two | Panama City (PA) | 246 | 73 | 2010 |
| 10 | Pearl Tower | Panama City (PA) | 242 | 70 | 2011 |

**Table 12.** Africa's 10 tallest skyscrapers in 2020.

| # | Building Name | City | Height (m) | Floors | Completion |
|---|---|---|---|---|---|
| 1 | Carlton Center | Johannesburg (ZA) | 223 | 50 | 1973 |
| 2 | Britam Tower | Nairobi (KE) | 200 | 33 | 2017 |
| 3 | Commercial Bank of Ethiopia | Addis Ababa (ET) | 198 | 46 | 2018 |
| 4 | The Leonardo | Sandton (ZA) | 188 | 48 | 2018 |
| 5 | Ponte Tower | Johannesburg (ZA) | 173 | 54 | 1975 |
| 6 | JW Marriott Hotel | Casablanca (MA) | 167 | 42 | 2017 |
| 7 | UAP Old Mutual Tower | Nairobi (KE) | 163 | 33 | 2016 |
| 8 | NECOM House | Lagos (NG) | 160 | 32 | 1979 |
| 9 | Abuja World Trade Center Hotel Tower | Abuja (NG) | 158 | 37 | - |
| 10 | PSPF Commercial Tower A | Dar es Salaam (TZ) | 153 | 35 | 2014 |

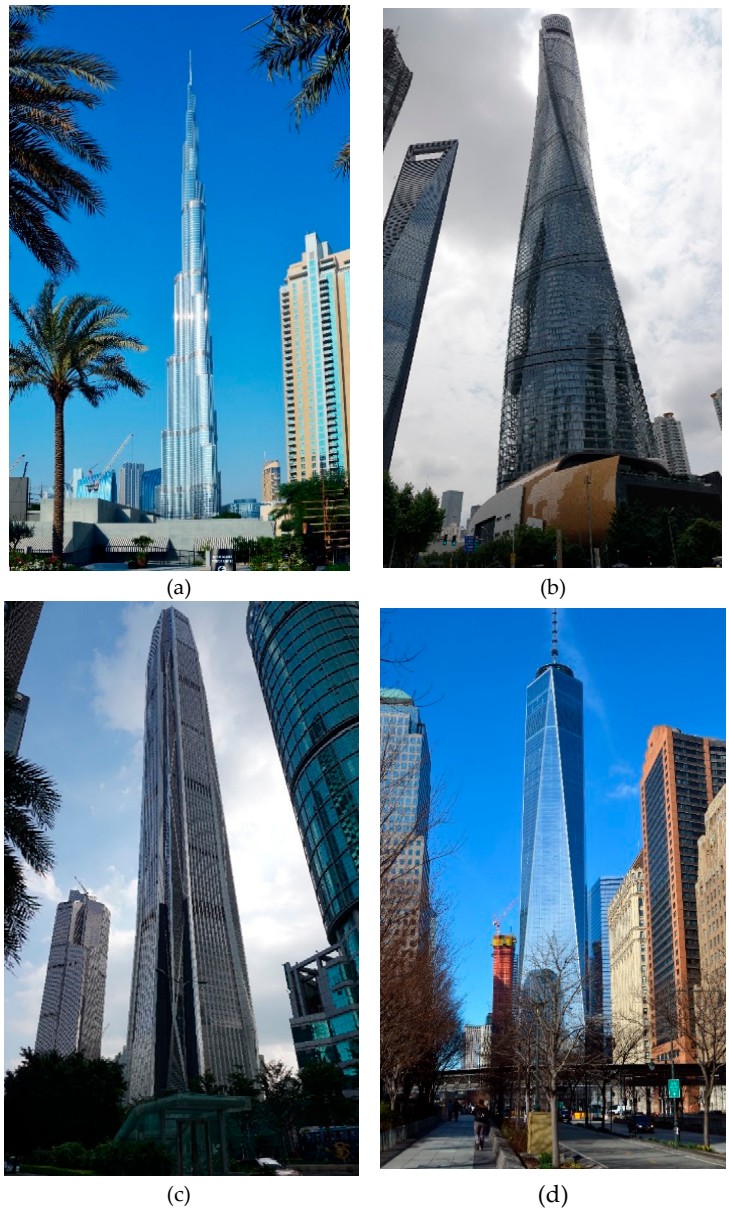

(a)　　　　　　　　　　　(b)

(c)　　　　　　　　　　　(d)

**Figure 1.** Examples of world's tallest buildings built in the 21st Century: (**a**) Burj Khalifa (828 m), (**b**) Shanghai Tower (632 m), (**c**) Ping An Finance Center (562 m), and (**d**) One World Trade Center (541 m), (*Photographs by author*).

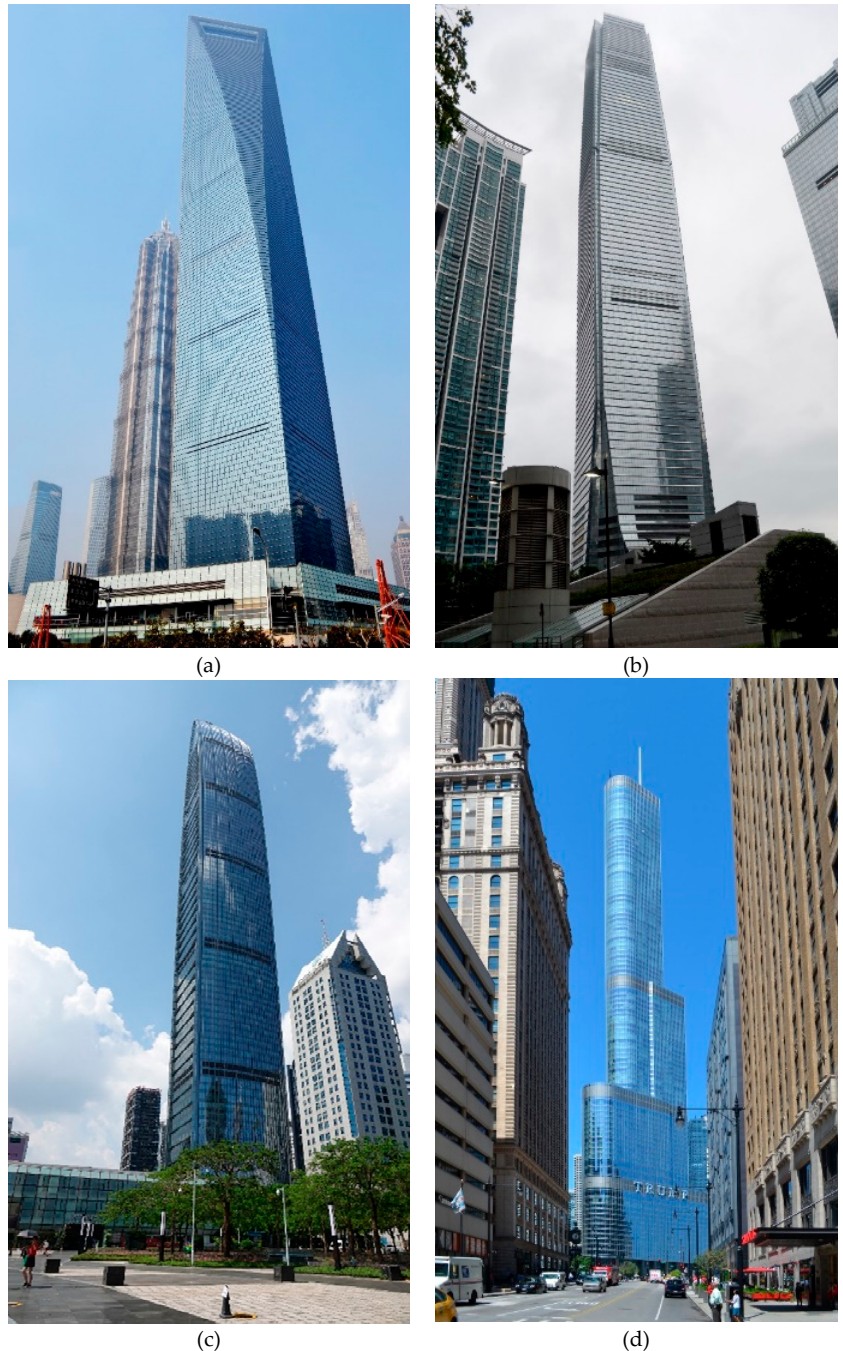

**Figure 2.** Examples of the world's tallest buildings built in the 21st Century: (**a**) Shanghai World Financial Center (492 m), (**b**) International Commerce Center (484 m), (**c**) KK 100 (441 m), and (**d**) Trump International Hotel & Tower (423 m) (Photographs by author).

## 2. Driving Forces

### 2.1. Population Increase and Migration

Among the pressing issues that have spurred tall building development, and will likely continue, is the exponential increase in urban population worldwide in conjunction with wealth accumulation. Currently, more than half of the world is urban when 20 years ago it was only one-third [16,17]. By 2030, it is expected that about 60% of the world's population will be urban. In 2050, about 75% of the world population will live in urban areas when the world's population is expected to reach 9 billion.

At that time, all major cities of the world especially those in Asia, Africa, and Latin America will have enormous populations likely ranging from 30 million to 50 million or more [18]. Accommodating such a large population in cities will be a colossal challenge. The horizontal scale of cities is continually being strained with no alternative but to build upward to accommodate city dwellers. Rural-to-urban migration is one of the causes of the urban population increase. In China, it is projected that, by 2025, 350 million people will migrate from a rural to an urban environment. Marcos Fava Neves predicted, "This will require five million buildings . . . equivalent of 10 cities the size of New York" [19]. In other words, Chinese cities need to build an equivalent amount to the U.S. population in merely 13 years to accommodate the population increase [19]. In such cases, high-rise development is almost certain to be part of the solution.

## 2.2. Demographic Change

Demographic shifts demonstrate that many of the millennials prefer living in urban centers that offer cultural amenities, vibrant social life, and mass transit. They prefer a car-free lifestyle enabled by having a workplace near home. The US housing market has expected that demand for single-family homes will continue to grow tremendously. Nonetheless, this did not occur. Instead, many people, particularly millennials and boomers, have decided to move into high-rises of central cities (e.g., New York City, Chicago, San Francisco, Atlanta, and Austin), seeking urban living. While millennials, or the Creative Class as described by the urban theorist Richard Florida, are finding "hot" jobs in the city, downsizing retirees are moving to walkable neighborhoods to free themselves from the maintenance burden of suburban homes. Both crowds, nevertheless, are enjoying urban amenities, services, mass transit, walkable places, and civic life offered in these city centers. In San Francisco, developers are building residential towers near Twitter's headquarters. Similarly, Seattle's developers are building residential towers near Amazon's global headquarters. Overall, "after 50 years of an immense suburban sprawl, there is a noteworthy flight back to cities, an increasing acceptance of density and demand on urban cores, as well as reemphasis on public transit as a key to a sustainable future," state Rebecca Leonard and Joe Porter [20] (p. 12).

## 2.3. Global Competition and Globalization

The ongoing trend for constructing tall buildings around the world reflects the increasing impact of global competition on the development of the world's major cities. These cities compete on the global stage to have the title of tallest building in which to announce the confidence and global stature of their growing economies. An iconic tall building enhances the global image of the city. It is likely to put the city on the world map and signal or promote its significant economic progress and advancement. Political leaders have supported constructing tall buildings to present their countries as emerging global economic powers. In some parts of the world, globalization has immensely promoted the local economy and, consequently, the construction of tall buildings. The City of Shenzhen, China, for example, was a small fishing village in the 1970s. Due to global forces and rapid foreign investment, it was transformed into a modern city of skyscrapers. Foreign nationals have invested billions of dollars in building factories and forming joint ventures. Today, the city is home to the headquarters of numerous high-tech companies that house their offices in major tall and supertall buildings [2].

## 2.4. Urban Regeneration

As explained earlier, city centers in developed countries that suffered from the migration of their population to the suburbs in the 1970–1990s have witnessed a major return to their centers in recent years. Therefore, cities are witnessing an urban renaissance and a desire to return to high-rise living in the urban cores. Tall buildings are viewed as tools to encourage central living and working. Construction of new attractive high-rises can also beautify and revitalize dilapidated districts and neighborhoods within the urban core and surrounding areas. This improves the quality of life in these areas by minimizing or eliminating social ills such as crime that might have been prevalent there [2].

### 2.5. Agglomeration

The tallness of buildings is also a matter of agglomeration in business districts. Urban agglomeration hinges on the proximity of activities and tall buildings do just that. Clustering tall buildings fosters urban synergy for diverse activities and specialized services. The high concentration of activities creates "knowledge spillovers" between firms in the same sector and across sectors that lead to increased innovation. In a denser and varied environment, knowledge can spill into unintended fields and a significant share of knowledge transfer occurs informally. David Audretsch explains: "Since knowledge is generated and transmitted more efficiently via local proximity, economic activity based on new knowledge has a high propensity to cluster within a geographic region . . . . Greater geographic concentration of production leads to more, and not less, dispersion of innovative activity . . . " [21] (p. 21). Clearly, the presence of a large concentration of firms offering similar products spurs competition, innovation, and efficiency. Agglomeration improves the economy of scale and can increase productivity through access to denser markets. Access to competing suppliers helps firms procure more efficient, cheaper, and more appropriate inputs. Researchers have attempted to quantify the impact of agglomeration. Colin Buchanan's research reveals that "a doubling of employment density within a given area can lead to a 12.5% additional increase in output per worker in that area. For the service sector, the figure is far higher at 22%" [22] (p. 590).

### 2.6. Land Prices

Land prices have been a prime driver for constructing tall buildings. A phrase for skyscrapers came from Cass Gilbert in 1900, "A skyscraper is a machine that makes the land pay" [23] (p. 71). In large cities, properties are expensive and buildings logically grow upward. Cheaper land keeps buildings closer to the ground. Tall buildings are not an attractive option for small towns where land is cheap [24]. Land prices recently have been significant drivers of tall building development in cities seeking to re-populate their urban centers with residential-recreational complexes inserted in the predominantly commercial-retail Central Business Districts (CBD). These relatively new markets drive up city center land prices, which make building upward for investment return increasingly necessary. In some cities such as London, Singapore, and Hong Kong land prices are very high, at about $30,000 per square meter. Therefore, developers maximize the site by building ultra-tall buildings between 50 to 80 floors [25]. In the case of New York City, Rem Koolhaas, in his book Delirious New York, explained that Manhattan has no choice but extruding the city grid vertically [26]. Similarly, in Mecca, Saudi Arabia, land nearby the Sacred Mosque (Al-Masjid Al-Haram) is limited and exceedingly expensive and, therefore, has recently witnessed significant high-rise development including the ultra-tall Abraj Al-Bait Towers.

### 2.7. Land Preservation

Sustainability promotes compact urban living and vertical density is viewed as a tool to create a more sustainable city. Urban planners and institutions such as the Urban Land Institute in the U.S. are supporting this view: "By strategically increasing the number of dwelling units per acre, cities not only will go a long way toward meeting their sustainability objectives but also will be competitive, resilient, and great places to live" [27] (p. 67). Dense arrangements help preserve open space—a core goal of sustainability—that aims at preserving different types of open spaces including natural areas in and around cities and localities that provide a habitat for plants and animals, recreational spaces, farm and ranch lands, places of natural beauty, critical environmental areas (e.g., wetlands), and recreational community spaces. The availability of open space provides significant environmental quality and health benefits that include improving air pollution, attenuating noise, controlling the wind, providing erosion control, and moderating temperatures. Open space also protects surface and ground water resources by filtering trash, debris, and chemical pollutants before they enter a water system. Through the "Towers-in-the-Park" model, Le Corbusier advocated the high-density city mainly for increasing

access to nature. For example, accommodating the same number of people in a tall building of 50 stories versus five stories requires about one-tenth of the land [28–30].

## 2.8. Climate Change and Energy Conservation

Global warming could change climate patterns, which makes droughts and excessive rainfall more frequent and severe. According to a NASA (National Aeronautics and Space Administration) study, the Arctic perennial sea ice has been decreasing at a rate of 9% per decade since the 1970s and is caused likely by climate change. These issues are significant for living conditions since they can profoundly affect our cities. For example, a 6 m (20 ft) rise in sea level would submerge all of South Florida [31]. Consequently, fighting global warming and reducing $CO_2$ emissions are becoming prime goals of countless cities. The Kyoto Protocol was created in 1997 to fight global warming and more than 180 states joined the protocol by 2009. In 2015, at the Paris climate conference (COP21), 195 countries adopted the "Paris Agreement," which is the first-ever universal, legally binding global climate treaty. Therefore, the increase in emissions will result in damaging the climate and, hence, the desperate need to stabilize carbon emission can hardly be overemphasized. In this context, tall buildings have the potential to consume less energy than low-rise complexes since they have several energy-effective attributes such as agglomeration, savings in auto fuel and travel time, and a reduction in losses in power lines [32]. The roof is a prime source of energy loss in a building in addition to the façade. As such, a 50-story building of 10 apartments per floor has one roof and 500 single-family homes with each having the same floor area of an apartment with 500 roofs. Energy loss from 500 roofs is greater than that from one roof [33]. Overall, vertical development not only reduces carbon emissions but also provides opportunities to create compact environments that feature efficient mobility and accessibility while offering a higher quality of life. "Compact, mixed use, walkable, transit-oriented places offer significant environmental, economic, and social benefits" [34].

## 2.9. Infrastructure and Transportation

The high cost of maintaining expansive infrastructure hurts taxpayers and contributes to the fiscal crisis that local governments face. The costs of providing and maintaining public infrastructure and services for a given community in a new sprawling development is higher than to service the same community in a "smart growth" or infill development. Largely, vertically configured buildings facilitate more efficient infrastructure. Simply put, a 500-unit single-family subdivision requires many more roads, sidewalks, sewers, hydro lines, power and gas lines, light standards, and fire hydrants than that of a tall building, which allows integrating these systems efficiently in a dense manner. A comprehensive review of dozens of studies published by the Urban Land Institute uncovered that since 1980, the number of miles Americans drive has grown three times faster than the population and almost twice as fast as vehicle registrations. The researchers conclude that one of the greatest ways to reduce carbon emission is to build compact places where people can accomplish more with less driving. Compact development reduces driving from 20% to 40% [35,36].

## 2.10. International Finance

Wealthy international people increasingly invest in the global economy and real estate. They find cities such as New York City and London to be safe places to park their money as opposed to keeping their wealth in their insecure and politically unstable home countries. International investors view tall and supertall buildings as excellent investments because they involve placing a huge amount of capital in one place. In these skyscrapers, international owners can choose to rent their residential units, live there temporary or permanently, or leave them vacant [2].

## 2.11. Air Rights

Buying "air right," also called transferable development rights (TDR), has become popular in dense cities in recent years due to high demand in areas that have fewer developable lots. Zoning

codes assign the greatest height for each zone and specify air right rules. The air right concept enables a low-rise building's owner to sell the air above a property (e.g., church, school, museum, apartment building) to a neighboring building, which can boost its allowable height. Consequently, the air above an "underbuilt" structure or "vertical" real estate space becomes a valuable commodity. Overall, the air right concept allows a neighborhood to tap into its "unused potential." New York City offers an illustrative example. NYC introduced the air right concept in 1961 and today, it witnesses a renewed demand [2]. For example, One57 (306 m/1004 ft) in Midtown Manhattan added to its height allotment by buying air rights from its neighboring buildings including the nine-story 140 West 58th Street built in 1930. Completed in 2014, the 92-story ultra-tall building offers splendid views of the nearby Central Park. Similarly, 220 Central Park South (nearing completion) will reach new heights by buying air rights from neighboring buildings. Remarkably, as land prices soar, air right prices soar as well [37,38].

### 2.12. Human Aspirations and Ego

According to Roberto Assagioli, a pioneer of psycho-synthesis theories, the conception of height has to do with "self-realization," "self-actualization," and "human potential." Consequently, humans have admired tall structures since ancient times. A commonality of the "seven wonders" of the ancient worlds (e.g., the Temple of Artemis at Ephesus, the Lighthouse of Alexandria, and the Great Pyramids of Egypt) is that all are tall and visible. Human spirit and resilience were the driving forces behind the skyscraper phenomenon that started in the late 19th century. Tall buildings can project a sense of socio-economic power and promote the city as a leading and modern commercial center. Skyscrapers epitomize people's pride in their cities and display the achievements of warm architectural passion and cold engineering logic [15]. Tall structures provide an identity for a city such as Big Ben and the Shard in London, the Eiffel Tower in Paris, the Space Needle in Seattle, Willis Tower in Chicago, Burj Khalifa in Dubai, and so on. Observation decks celebrate human ascendance over the sky and the surrounding landscape by providing unique panoramic views of the world below. Humanity has a preoccupation with building large and tall constructs to defy gravity. "Tall has power." Imagining modern cities without skyscrapers is antithetical to the human spirit, pride, and identity. Therefore, human ego has a role in building skyscrapers.

### 2.13. Emerging Technologies

The evolution of the tall building has led to major advancements in engineering and technology. As today's technology becomes increasingly sophisticated, architects have an opportunity to build taller and implement their desire for the latest and utmost aesthetic expressions of tall buildings. Simultaneously, the desire for achieving higher quality tall buildings has encouraged research in areas such as building service systems, computer sciences, façade engineering, glazing, daylight and heat control, structural framing systems, ceiling systems, lighting, ventilation, exit strategies, water recycling systems, and more. Tall buildings have challenged technology itself and allowed to build towers more efficiently and sustainably and to create internal environments that are comfortable, productive, and energy-efficient. The prevalent green movement has propelled the design of high-performance tall buildings by employing intelligent technologies and smart materials.

## 3. Global Cities

### 3.1. Asia

#### 3.1.1. Shanghai

As a major global city, Shanghai has been among the most active cities in constructing tall buildings. Overall, in the first two decades of the 21st Century, Shanghai will add 156 skyscrapers to the 69 skyscrapers it built previously. Furthermore, 63 of these skyscrapers are in the Lujiazui Financial District. About 40 skyscrapers are located immediately west of the district across the Huangpu River

and the rest proliferates throughout the city. In addition, 37 of the added 47 200 m+ skyscrapers are located in the Lujiazui district and the rest are placed immediately west of the district across the river. Three of the five added supertalls (Shanghai Tower, Shanghai World Financial Center, and Jin Mao) are also located in the Lujiazui district, which creates a strong focal point in the city's skyline [39]. The average height of the 10 tallest skyscrapers in 2000 and 2020 will increase from 235 m to 365 m, equaling a 160% increase (Table 13). Certainly, Shanghai is one of the most active cities in building significant skyscrapers and it has been transformed from a polycentric city to a super polycentric city (Figure 3).

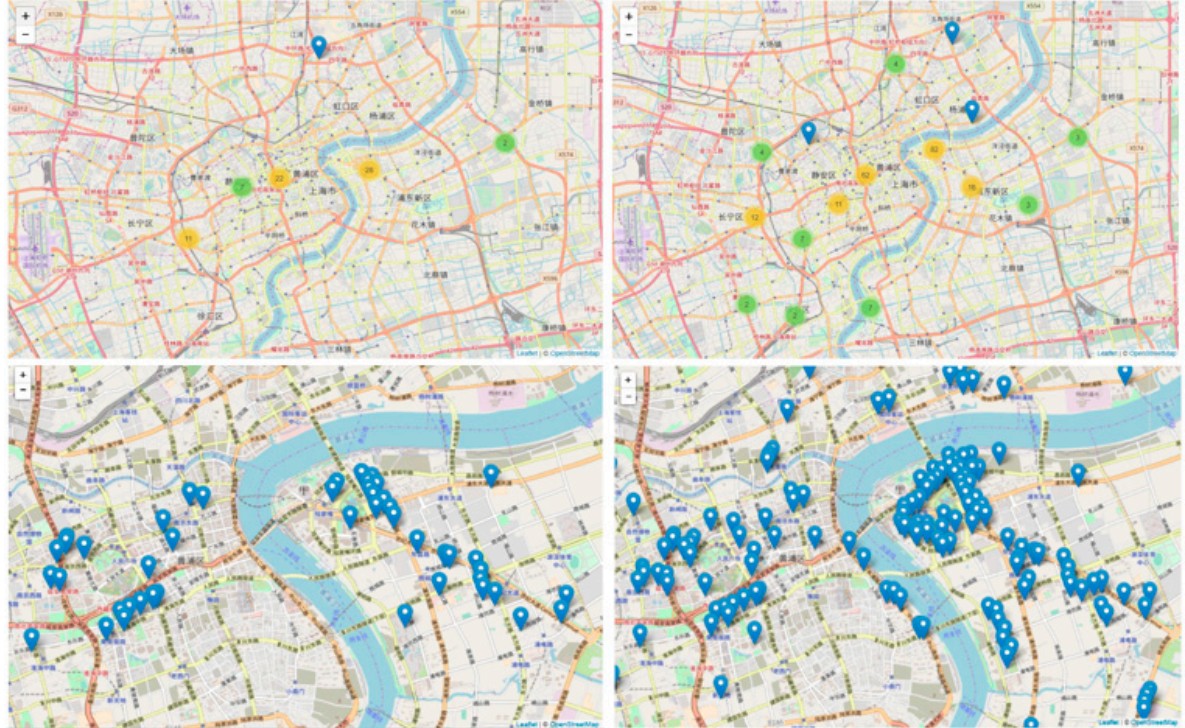

**Figure 3.** Shanghai in 2000 and 2020.

**Table 13.** Shanghai's 10 tallest skyscrapers in 2000 and 2020.

| # | Building Name | Height (m) | Building Name | Height (m) |
|---|---|---|---|---|
| 1 | Jin Mao Tower | 420.5 | Shanghai Tower | 632 |
| 2 | International Ocean Shipping Building | 232.4 | Shanghai World Financial Center | 492 |
| 3 | BOCOM Financial Towers | 230.4 | Jin Mao Tower | 420.5 |
| 4 | Bank of China Tower | 226.1 | Shimao International Plaza | 333.3 |
| 5 | World Finance Tower | 212 | Zhenru Center | 330 |
| 6 | Xinjinqiao Mansion | 212 | Sinar Mas Center 1 | 319.6 |
| 7 | Golden Bell Mansion | 208 | Plaza 66 | 288.2 |
| 8 | Nan Zheng Mansion | 205 | Tomorrow Square | 284.6 |
| 9 | Lippo Plaza | 204 | Hong Kong New World Tower | 278.3 |
| 10 | HSBC Tower | 203.4 | Shanghai Wheelock Square | 270.5 |

### 3.1.2. Beijing

Beijing, China's capital, has recently engaged in a race with its rival city, Shanghai, in building significant tall buildings. Scheduled for completion in 2018, China Zun Tower will be the flagship building of a 30-hectare master plan for the central business district. Twenty tall buildings ranging from 150 m to 350 m in height will cluster around Zun Tower [8,39]. China Zun Tower will rise to 528 m (1732 ft) and will become the tallest building in Beijing and the eighth tallest in China. Overall, in the first two decades of the 21st century, Beijing will add 55 skyscrapers to the 10 skyscrapers it built previously. Almost all of these buildings are located in close proximity (within a square mile) in

the east side of the city center. Eight of the 200 m+ buildings are located in this area including one supertall building (Figure 4). The average height of the 10 tallest skyscrapers in 2000 and 2020 will increase from 147 m to 291 m, equaling a 200% increase (Table 14).

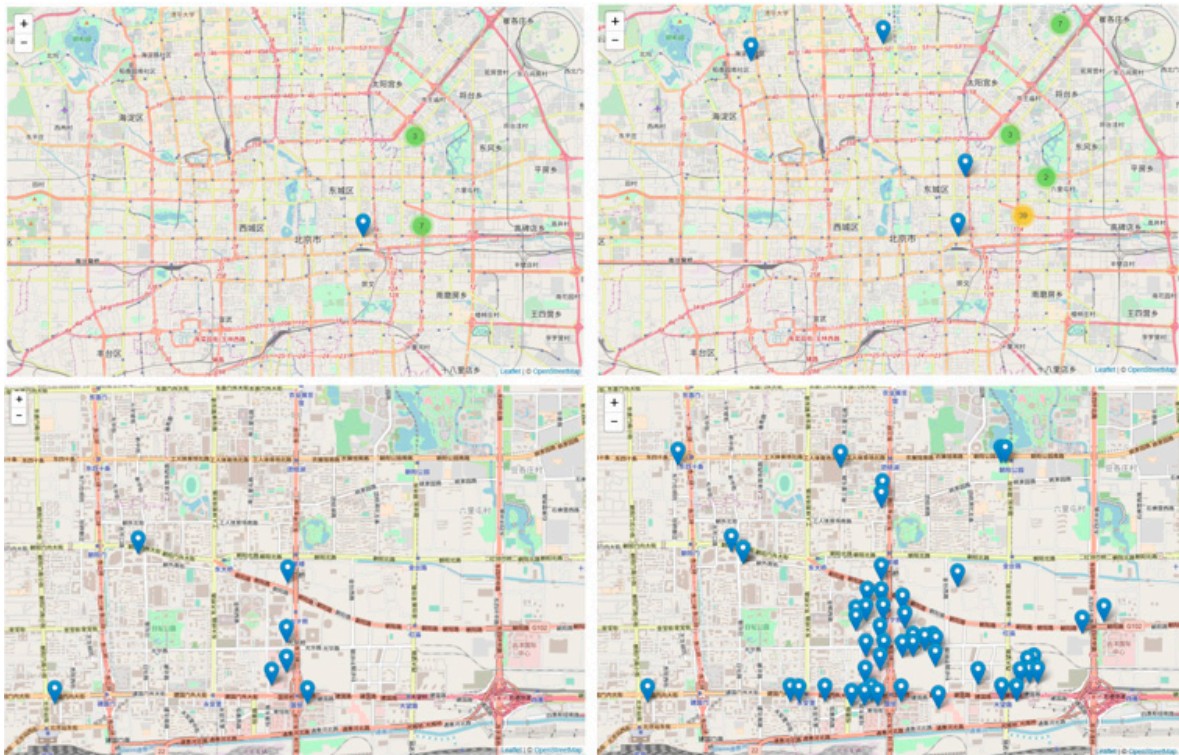

**Figure 4.** Beijing in 2000 and 2020.

**Table 14.** Beijing's 10 tallest skyscrapers in 2000 and 2020.

| # | Building Name | Height (m) | Building Name | Height (m) |
|---|---|---|---|---|
| 1 | Jing Guang Center | 208 | Citic Tower | 528 |
| 2 | Capital Mansion | 183 | China World Tower | 330 |
| 3 | China World Trade Center I | 155 | China World Trade Center Phase 3B | 295.6 |
| 4 | China World Trade Center II | 155 | Fortune Financial Center | 267 |
| 5 | China Merchants Group Tower | 148.1 | Samsung China Headquarters | 260 |
| 6 | Beijing Silver Tower | 145 | Beijing Greenland Center | 260 |
| 7 | China Life Tower | 130.8 | Beijing Yintai Center—Park Tower | 250 |
| 8 | Beijing Kerry Centre Office Building | 123.5 | Sunshine Insurance Headquarters | 243 |
| 9 | Henderson Center | 110 | Beijing Television Center | 239 |
| 10 | Henderson Center | 110 | Z14 Plot Tower 1 & 2 | 238 |

### 3.1.3. Shenzhen

An important emerging global skyscraper city in China is Shenzhen. Since becoming China's first Special Economic Zone, Shenzhen has witnessed an unprecedented urban growth and expanded from 300,000-people to more than 10 million-city in 35 years. Shenzhen has built significant skyscrapers. Located in the center of Shenzhen and completed recently, Ping An Finance Center (PAFC) rises to 599 m (1965 ft), which ranks as the second tallest building in China after Shanghai Tower. Overall, in the first two decades of the 21st Century, Shenzhen will add 172 skyscrapers to the 20 skyscrapers it built previously. This sudden and massive increase in skyscrapers has not only reinforced existing modest clusters but created totally new ones [13,39,40]. The locations of the new 79 200 m+ skyscrapers stretch across the two far ends of the city, east and west (Figure 5). The locations of the newly constructed five supertalls reinforce "downtown" clusters. The average height of the 10 tallest skyscrapers in 2000 and 2020 will increase from 231 m to 393 m, equaling a 170% increase (Table 15).

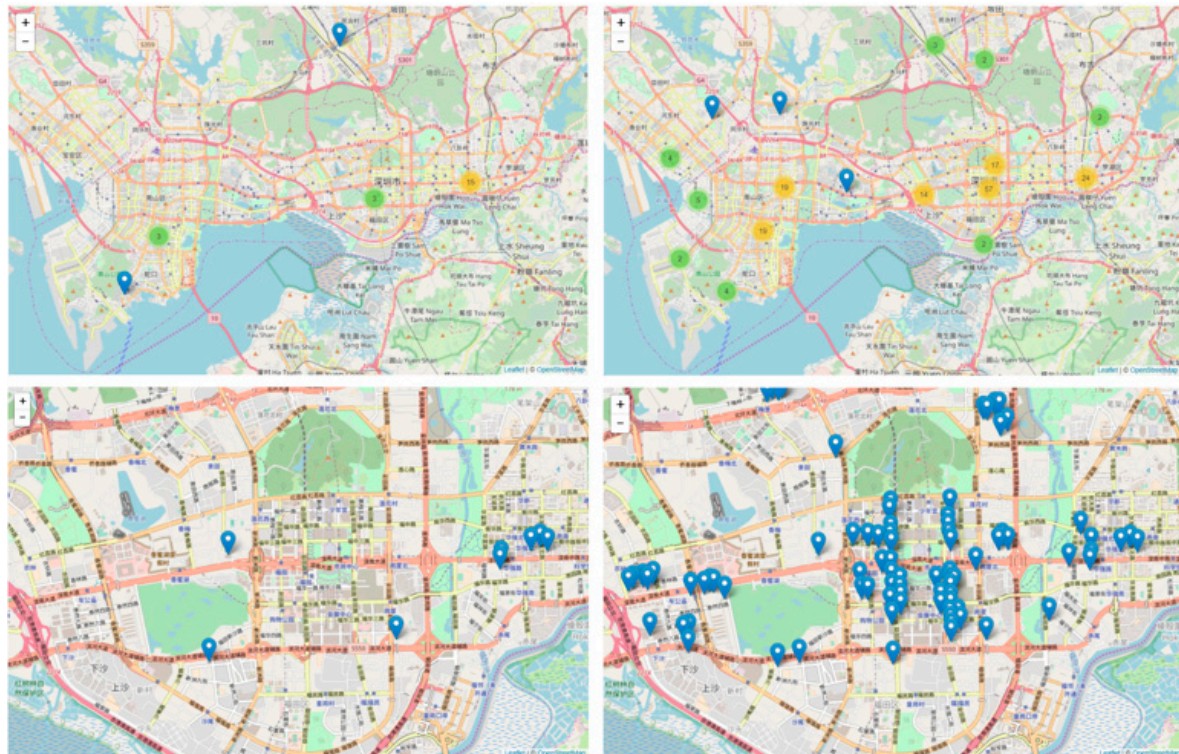

**Figure 5.** Shenzhen in 2000 and 2020.

**Table 15.** Shenzhen's 10 tallest skyscrapers in 2000 and 2020.

| # | Building Name | Height (m) | Building Name | Height (m) |
|---|---|---|---|---|
| 1 | Shun Hing Square | 384 | Ping An Finance Center | 599.1 |
| 2 | SEG Plaza | 291.6 | KK100 | 441.8 |
| 3 | Shenzhen Special Zone Daily Tower | 260 | China Resources Tower | 392.5 |
| 4 | Panglin Plaza | 240 | Shum Yip Upperhills Tower 1 | 388.1 |
| 5 | Guotong Building | 201 | Shun Hing Square | 384 |
| 6 | United Plaza A | 195 | Shenzhen Center | 375.6 |
| 7 | New Era Plaza | 188.7 | Hanking Center Tower | 350 |
| 8 | Shenzhen Development Bank | 183.6 | One Shenzhen Bay Tower 7 | 341.4 |
| 9 | Shenzhen International Science and Technology Building | 182 | Shimao Qianhai Project Tower 1 | 330 |
| 10 | Electronics Science & Technology Building | 181 | Hon Kwok City Center | 329.4 |

### 3.1.4. Bangkok

Bangkok, the capital of Thailand, is also an emerging skyscraper city in Asia. Recently, Bangkok has completed its tallest building, which is the 314-m (1031-ft) MahaNakhon. The Magnolias Waterfront Residences Tower 1, however, will snatch the tallest building title from MahaNakhon soon by rising higher by merely one meter. Other important towers under construction include Capella Residences (300 m (983 ft)), Magnolias Waterfront Residences Tower 2 (280 m (919 ft)), and Canapaya Residences I (253 m (830 ft)). Lately, SOM has unveiled a vision for a 16-hectare "skyscraper village" in the city [2,14]. Named "One Bangkok", it will house more than 60,000 people and integrate a variety of parks and plazas, which will total eight hectares. Overall, in the first two decades of the 21st Century, Bangkok will add 95 skyscrapers to the 27 skyscrapers it built previously [2]. The new buildings are located in close proximity to older ones, which reinforce the image of the city's downtown (Figure 6). The average height of the 10 tallest skyscrapers in 2000 and 2020 will increase from 205 m to 274 m, equaling a 134% increase (Table 16).

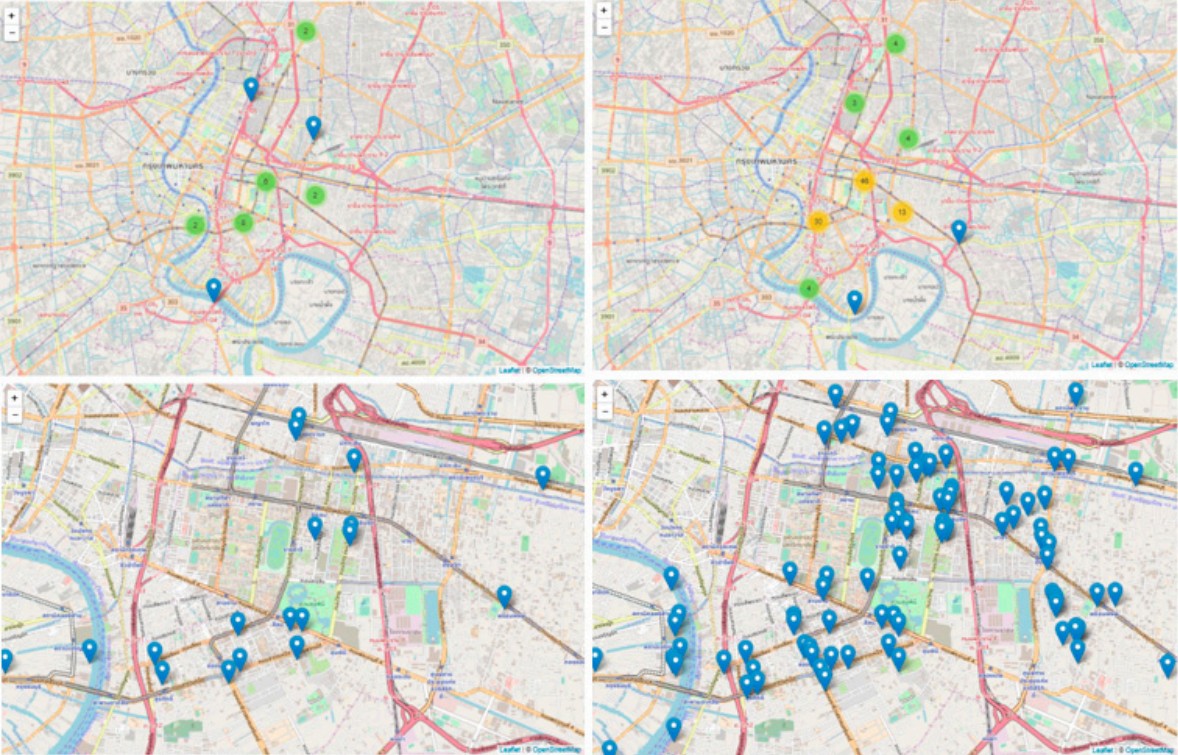

**Figure 6.** Bangkok in 2000 and 2020.

**Table 16.** Bangkok's 10 tallest skyscrapers in 2000 and 2020.

| # | Building Name | Height (m) | Building Name | Height (m) |
|---|---|---|---|---|
| 1 | Baiyoke Tower II | 304 | Magnolias Waterfront Residences Tower 1 | 315 |
| 2 | Empire Tower | 226.8 | MahaNakhon | 314.2 |
| 3 | Jewelry Trade Center | 220.7 | Baiyoke Tower II | 304 |
| 4 | Sinn Sathorn Tower | 195 | Four Seasons Hotel and Private Residences and Capella Hotel | 299.5 |
| 5 | Thai Wah Tower II | 194 | Mandarin Oriental Residences Bangkok | 268.7 |
| 6 | United Center | 187 | The River South Tower | 258 |
| 7 | Abdulrahim Place | 187 | Canapaya Residences I | 253 |
| 8 | Tipco Tower | 180 | State Tower | 247.2 |
| 9 | Kasikorn Bank Head Office | 177 | Waldorf Astoria + Magnolias Ratchaprasong | 242 |
| 10 | Sathorn House | 174.2 | Eastin Grand Hotel Phayathai Bangkok | 239.7 |

## 3.2. Europe

### 3.2.1. London

As a global financial city, London has been facing an increased demand for commercial office space. Consequently, it has been building significant skyscrapers [41]. Despite the recent Brexit fears and continuing objection by local communities, tall building construction in London continues with full force. In his article "London's Future Skyline: the 455 New Skyscrapers Turning our Capital into Manhattan," Patrick Scott (2017) explains, "More tall buildings were constructed in London last year than ever before with 26 opening their doors and 455 more in the pipeline, according to new industry research" [42]. The Shard (306 m (1004 ft)) in 2013, 110 Bishopsgate (230 m (755 ft)) in 2011, Leadenhall Building (224 m (735 ft)) in 2014, and 20 Fenchurch (160 m (525 ft)) in 2014 are among the significantly tall buildings developed recently in London [8,43]. Overall, in the first two decades of the 21st century, London will add 80 skyscrapers to the 23 skyscrapers it built previously. Furthermore, 21 of these new skyscrapers are located in Canary Warf, a new mixed-use area east of London's downtown, while the rest are scattered in the downtown north and south of the Thomas River (Figure 7). London added only eight 200 m+ skyscrapers including four in Canary Wharf and four in the downtown with only

one of a height greater than 300 m. The average height of the 10 tallest skyscrapers in 2000 and 2020 will increase from 142 m to 281 m, equaling a 198% increase (Table 17).

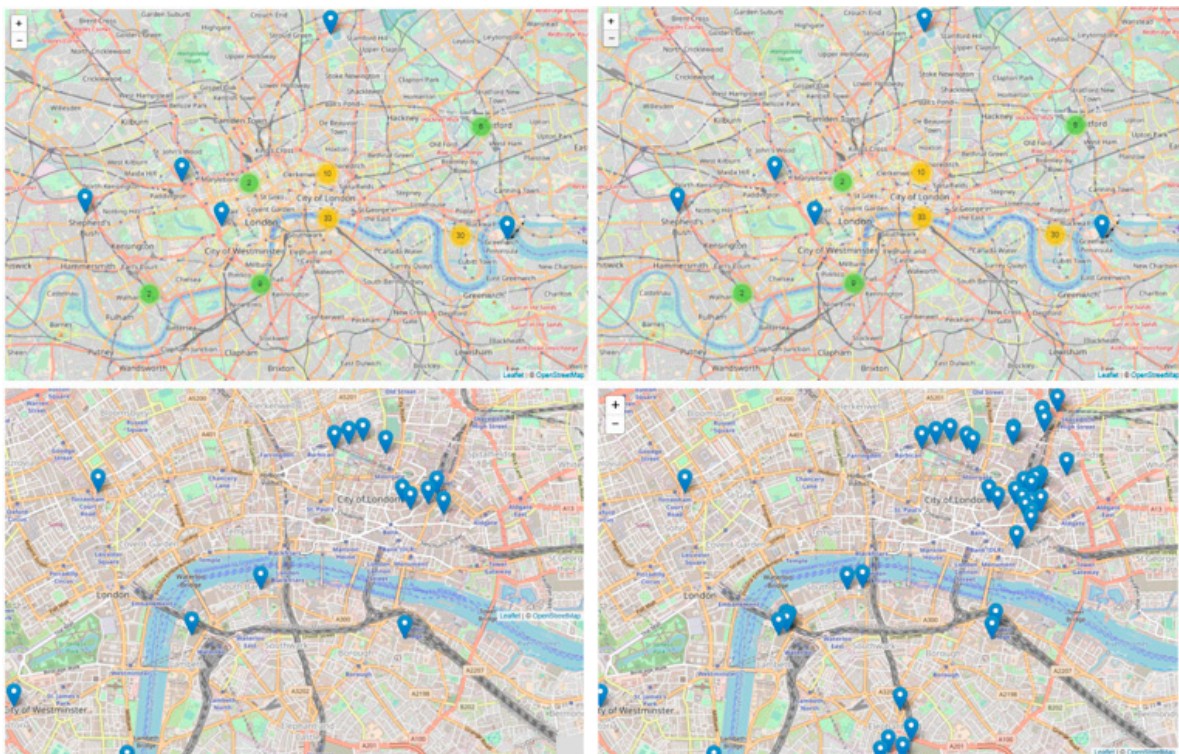

**Figure 7.** London in 2000 and 2020.

**Table 17.** London's 10 tallest skyscrapers in 2000 and 2020.

| # | Building Name | Height (m) | Building Name | Height (m) |
|---|---|---|---|---|
| 1 | One Canada Square | 236 | The Shard | 306 |
| 2 | Tower 42 | 182.9 | Twenty-two | 278.2 |
| 3 | Guy's Tower | 143 | One Canada Square | 236 |
| 4 | CityPoint | 127.1 | Landmark Pinnacle | 233.2 |
| 5 | Euston Tower | 124.3 | Salesforce Tower | 230 |
| 6 | Cromwell Tower | 123 | The Leadenhall Building | 224 |
| 7 | Lauderdale Tower | 123 | Newfoundland | 219.8 |
| 8 | Shakespeare Tower | 123 | Valiant Tower | 214.5 |
| 9 | Millbank Tower | 118.9 | One Park Drive | 204.9 |
| 10 | Aviva Tower | 117.9 | 25 Canada Square | 659 |

### 3.2.2. Moscow

Among all Russian cities, Moscow has been very active in constructing tall buildings. New buildings were spread out in multiple centers, which reinforce the polycentric nature of the city. Remarkably, Moscow International Business Center (also known as Moscow City) has some of Europe's tallest buildings with the following ranks: first, second, third, fifth, sixth, and seventh [3]. When completed in 2016, Federation Towers—Vostok Tower (374 m/1227 feet) became the tallest building in Russia and Europe. Overall, in the first two decades of the 21st Century, Moscow will add 151 skyscrapers to the 31 skyscrapers it built previously (Figure 8). The average height of the 10 tallest skyscrapers in 2000 and 2020 will increase from 164 m to 314 m, equaling a 191% increase (Table 18).

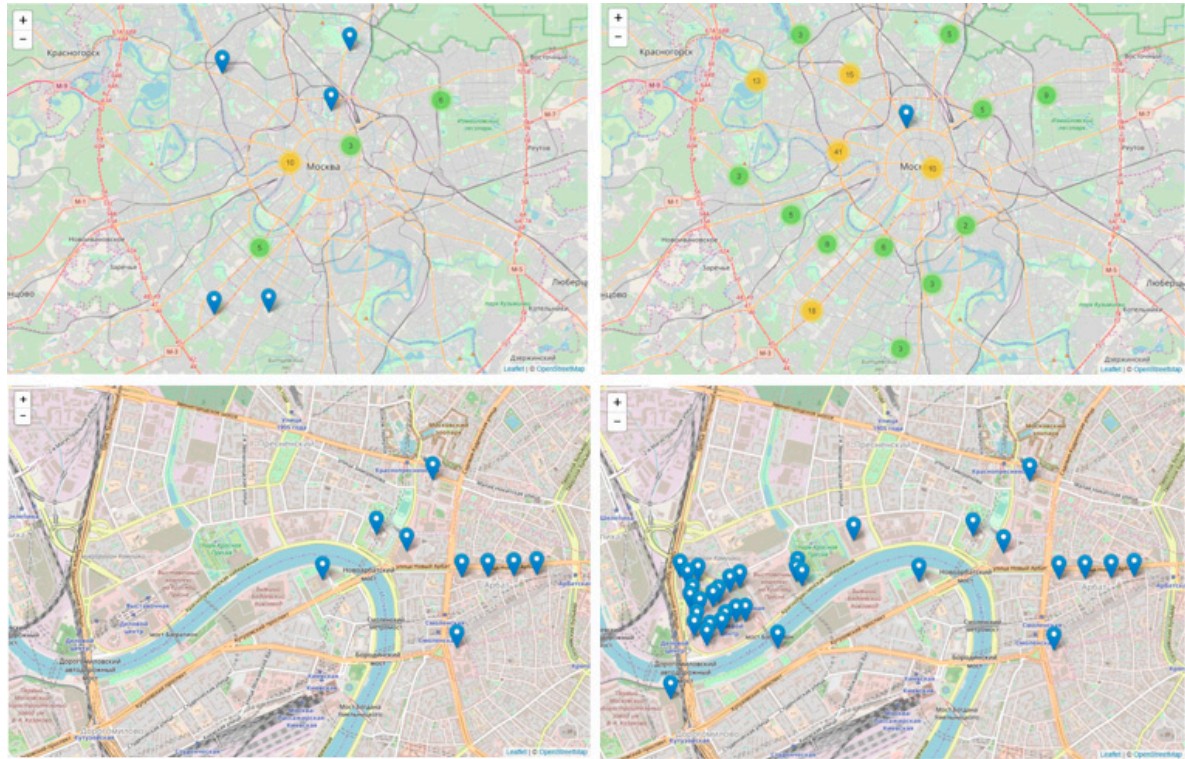

**Figure 8.** Moscow in 2000 and 2020.

**Table 18.** Moscow's 10 tallest skyscrapers in 2000 and 2020.

| # | Building Name | Height (m) | Building Name | Height (m) |
|---|---|---|---|---|
| 1 | MV Lomonosov State University | 239 | Federation Tower | 373.7 |
| 2 | Radisson Royal Hotel | 206 | OKO—Residential Tower | 353.6 |
| 3 | Kotelnicheskaya Naberezhnaya | 176 | NEVA TOWERS 2 | 345 |
| 4 | Ministry of Foreign Affairs | 172 | Mercury City Tower | 338.8 |
| 5 | Kudrinskaya Square | 156 | Stalnaya Vershina | 308.9 |
| 6 | Gazprom | 151 | Capital City Moscow Tower | 301.8 |
| 7 | Central Tourist House | 138 | NEVA TOWERS 1 | 297 |
| 8 | Red Gate Square | 138 | Grand Tower | 283 |
| 9 | Hilton Moscow Leningradskaya | 136 | Naberezhnaya Tower Block C | 268.4 |
| 10 | NII Delta Business Center | 132 | Capital Tower 1 | 267 |

*3.3. North America*

3.3.1. New York City

Notably, New York City experienced a rapid pace of building exceptional high-rises. From 1930 to 2015, the city completed 56 200 m+ (656 ft+) buildings while, in the third quarter of 2015, the city had 25 200 m+ buildings under construction, which is almost half the figure that it took 85 years to complete [44,45]. Similarly, between 1930 and 2015, the city built eight supertalls (300 m+). There are currently 14 such buildings under construction or proposed to rise by 2020. Particularly, Manhattan is likely to build more than 30 200 m+ buildings in the next five to 10 years, as compared to merely four since 2010 [43]. Currently, NYC holds 32% of all North America's 200 m+ buildings (63/196) and will have 35% (90/255) by 2020. Furthermore, NYC holds six out of the 10 tallest skyscrapers in North America and it will sustain this rank in 2020. Overall, NYC "remains the world's skyscraper capital and a long-time established center of global business," as Richard Barkham and colleagues explain in their recent article (2017) titled "Reaching for the Sky: The Determinants of Tall Office Development in Global Gateway Cities" [44] (p. 24). Overall, in the first two decades of the 21st Century, New York City

will add 304 skyscrapers to the 590 skyscrapers it built previously. Most of these buildings reinforce existing clusters in Midtown, Lower Manhattan and along Central Park [20]. The city's new 38 200 m+ skyscrapers reinforce existing clusters in which 20 are located in Midtown and 16 in Lower Manhattan. Similarly, all new eight 300 m+ skyscrapers reinforce these two clusters, six in Midtown and two in Lower Manhattan (Figure 9). The average height of the 10 tallest skyscrapers in 2000 and 2020 will increase from 314 m to 398 m, equaling a 127% increase (Table 19).

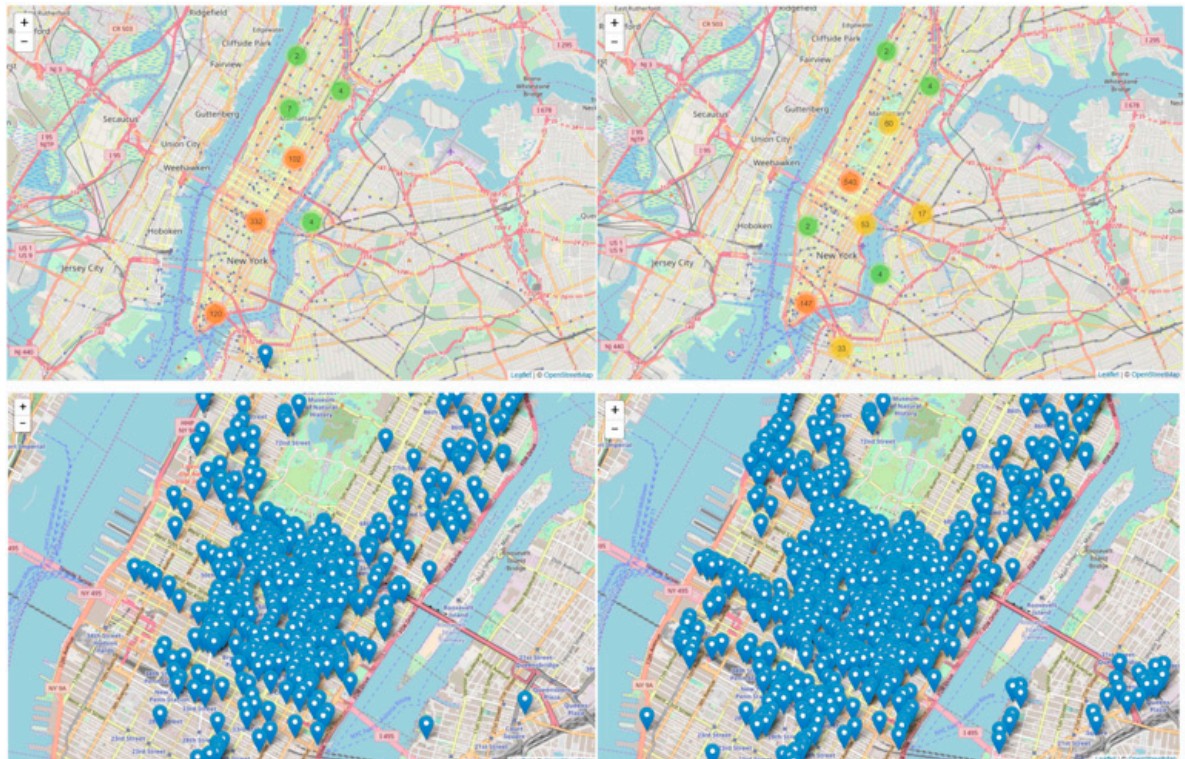

**Figure 9.** New York City in 2000 and 2020.

**Table 19.** New York City's 10 tallest skyscrapers in 2000 and 2020.

| # | Building Name | Height (m) | Building Name | Height (m) |
|---|---|---|---|---|
| 1 | One World Trade Center | 417 | One World Trade Center | 541.3 |
| 2 | Two World Trade Center | 415.1 | Central Park Tower | 472.4 |
| 3 | Empire State Building | 381 | 111 West 57th Street | 435.3 |
| 4 | Chrysler Building | 318.9 | 432 Park Avenue | 425.7 |
| 5 | 70 Pine | 290.2 | 30 Hudson Yards | 386.6 |
| 6 | The Trump Building | 282.6 | Empire State Building | 381 |
| 7 | 601 Lexington | 278.9 | Bank of America Tower | 365.8 |
| 8 | Comcast Building | 259.1 | 3 World Trade Center | 328.9 |
| 9 | CitySpire | 248.1 | 53 West 53rd | 320 |
| 10 | 28 Liberty | 247.8 | Chrysler Building | 318.9 |

### 3.3.2. Chicago

Although Chicago and the State of Illinois at large have been losing their population rates in the past two decades, downtown Chicago has been gaining population and building new skyscrapers. City planners focus on reviving the Loop Community (Chicago's CBD) because they view it to be critical for the city's long-term sustainable economic growth. Salient examples of skyscrapers (under construction and completed) include Studio Gang's 98-story Vista Tower, Rafael Viñoly's 76-story One Grant Park, HPA's 56-story Essex on the Park, Robert M. Stern's 67-story One Bennett Park, Pickard Chilton's 52-story River Point and Goettsch Partners' 53-story 150 North Riverside [3]. Overall, in the

first two decades of the 21st Century, Chicago will add 117 skyscrapers to the 207 skyscrapers it built previously. Most of these buildings reinforce existing clusters such as the Chicago Loop (Figure 10). The average height of the 10 tallest skyscrapers in 2000 and 2020 will increase from 308 m to 326 m, equaling a 106% increase (Table 20).

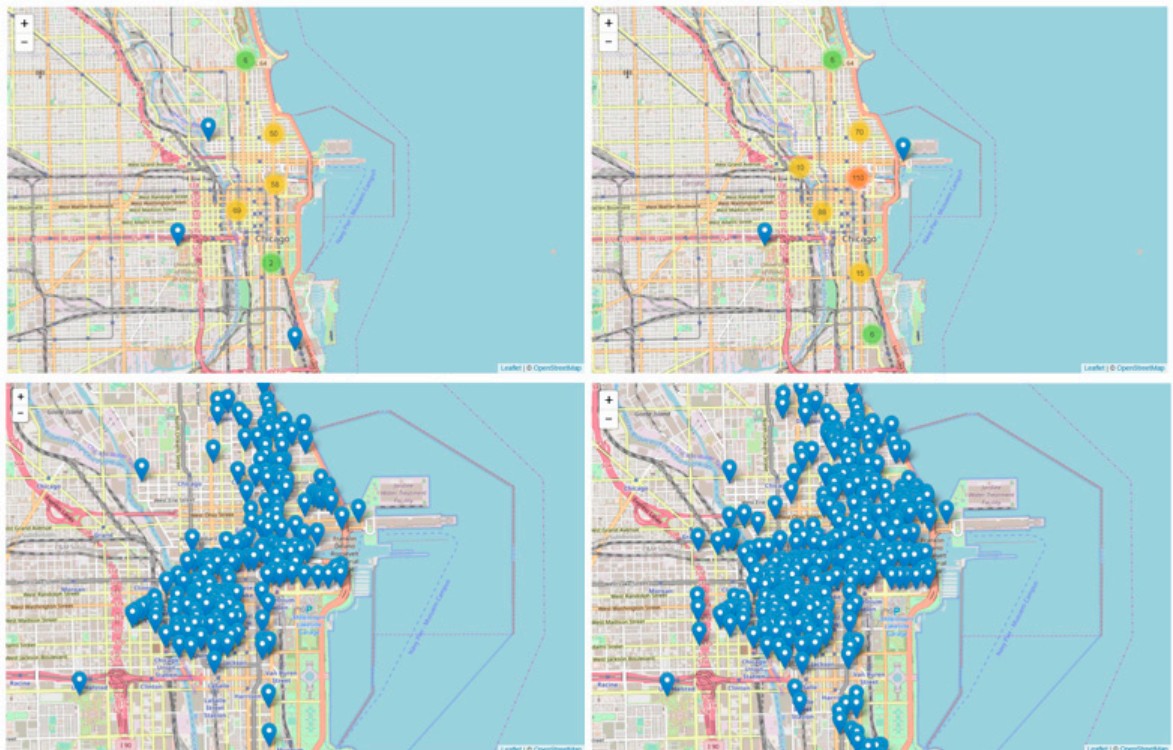

**Figure 10.** Chicago in 2000 and 2020.

**Table 20.** Chicago's 10 tallest skyscrapers in 2000 and 2020.

| # | Building Name | Height (m) | Building Name | Height (m) |
|---|---|---|---|---|
| 1 | Willis Tower | 442.1 | Willis Tower | 442.1 |
| 2 | Aon Center | 346.3 | Trump International Hotel & Tower | 423.2 |
| 3 | 875 North Michigan Avenue | 343.7 | Vista Tower | 362.9 |
| 4 | The Franklin—North Tower | 306.9 | 875 North Michigan Avenue | 343.7 |
| 5 | Two Prudential Plaza | 303.3 | The Franklin—North Tower | 306.9 |
| 6 | 311 South Wacker Drive | 292.9 | Two Prudential Plaza | 303.3 |
| 7 | 900 North Michigan Avenue | 265 | 311 South Wacker Drive | 292.9 |
| 8 | Chase Tower | 264.6 | NEMA Chicago | 270.4 |
| 9 | Water Tower Place | 261.9 | 900 North Michigan Avenue | 265 |
| 10 | Park Tower | 257.4 | Chase Tower | 264.6 |

### 3.3.3. Miami

Miami has been experiencing a boom in high-rise construction. Among the significant towers rising in the city are Panorama Tower, Brickell Flatiron, One Thousand Museum, Miami Worldcenter, and Grove at Grand Bay. Soaring to 252 m (828 ft), Panorama Tower has recently snatched the city's tallest building title from the 240 m (789 ft) Four Seasons Hotel & Tower. Overall, in the first two decades of the 21st Century, Miami will add 93 skyscrapers to the 27 skyscrapers it built previously. All new skyscrapers reinforce existing skyscraper clusters in the downtown and along the Atlantic shore (Figure 11). Many of the new skyscrapers fill spatial gaps, which creates a stronger continuous skyline. Three of the five 200 m+ skyscrapers are located in the southern part of the downtown and two are in the upper part. Before and after 2000, the city has no supertall buildings [2]. The average

height of the 10 tallest skyscrapers in 2000 and 2020 will increase from 154 m to 219 m, equaling a 142% increase (Table 21).

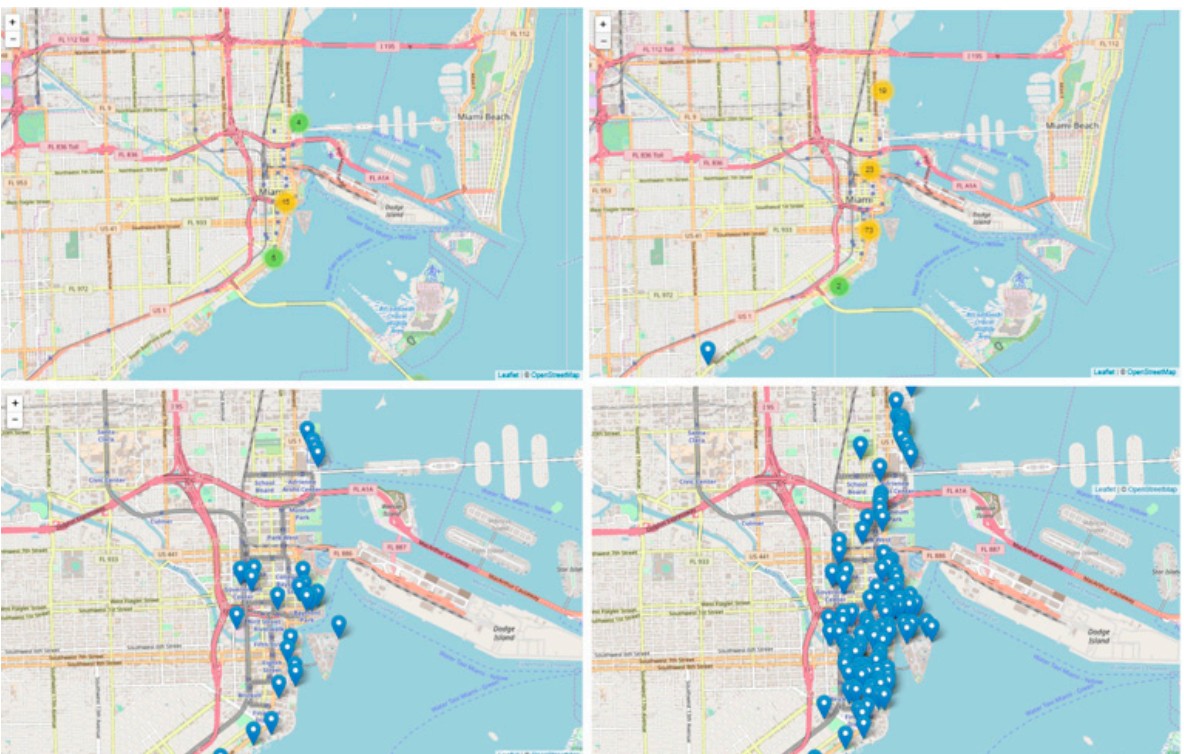

**Figure 11.** Miami in 2000 and 2020.

**Table 21.** Miami's 10 tallest skyscrapers in 2000 and 2020.

| # | Building Name | Height (m) | Building Name | Height (m) |
|---|---|---|---|---|
| 1 | Southeast Financial Center | 232.8 | Panorama Tower | 252.4 |
| 2 | Miami Tower | 190.5 | Four Seasons Hotel & Tower | 240.4 |
| 3 | Santa Maria | 158.5 | Southeast Financial Center | 232.8 |
| 4 | Stephen P. Clark Center | 155.4 | Brickell Flatiron | 224.3 |
| 5 | One Biscayne Tower | 150 | Marquis | 214 |
| 6 | Miami Center | 147.5 | One Thousand Museum | 213.1 |
| 7 | 701 Brickell Avenue | 137.1 | Paramount Miami World Center | 213.1 |
| 8 | Two Tequesta Point | 125 | Wells Fargo Tower | 199.5 |
| 9 | Courthouse Center | 123.4 | 900 Biscayne Bay | 198 |
| 10 | The Palace | 121.9 | Elysee | 197.8 |

### 3.3.4. San Francisco

San Francisco is building significant high-rises near the Transbay Transit Center, e.g., Oceanwide Center by Foster + Partners and Heller Manus, and Waldorf Astoria Hotel. Transbay Transit Center has three important buildings including Salesforce Tower, 181 Fremont, and Transbay Center Residential Tower. Recently completed, Salesforce Tower (326 m (1070 ft)) has become the city's tallest building. Overall, in the first two decades of the 21st Century, San Francisco will add 29 skyscrapers to the 67 skyscrapers it built previously [2]. All these buildings reinforce the current skyscraper cluster in the downtown area and they are located within one square mile (Figure 12). They are also in close proximity to major mass transit services. In addition, the city's only two 200 m+ skyscrapers including the 300 m+ Salesforce Tower are located at the epicenters of these transit areas. The average height of the 10 tallest skyscrapers in 2000 and 2020 will increase from 194 m to 221 m, equaling a 114% increase (Table 22).

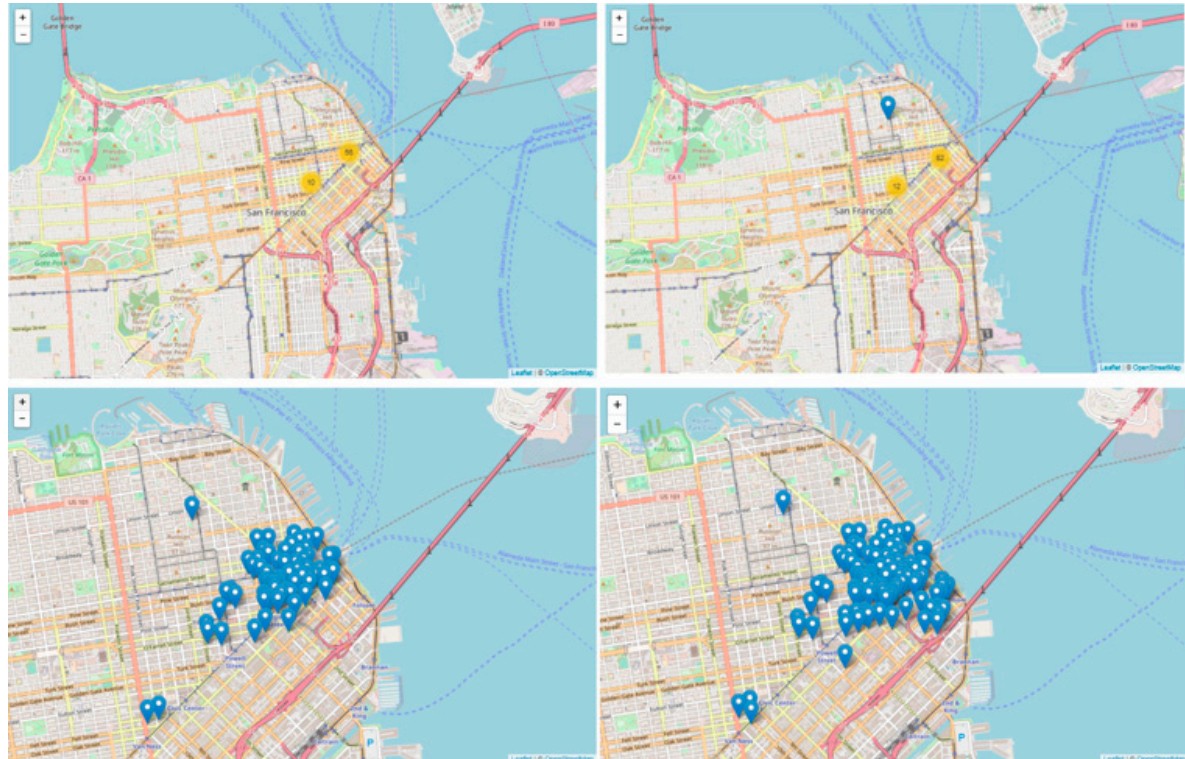

**Figure 12.** San Francisco in 2000 and 2020.

**Table 22.** San Francisco's 10 tallest skyscrapers in 2000 and 2020.

| # | Building Name | Height (m) | Building Name | Height (m) |
|---|---|---|---|---|
| 1 | Transamerica Pyramid Center | 260 | Salesforce Tower | 326.1 |
| 2 | 555 California Street | 237.4 | Transamerica Pyramid Center | 260 |
| 3 | 345 California Center | 211.8 | 181 Fremont | 244.5 |
| 4 | 101 California Street | 183 | 555 California Street | 237.4 |
| 5 | 50 Fremont Center | 183 | 345 California Center | 211.8 |
| 6 | 575 Market Street | 174.7 | Millennium Tower | 196.6 |
| 7 | Four Embarcadero Center | 173.7 | Park Tower at Transbay | 184.5 |
| 8 | One Embarcadero Center | 173.4 | One Rincon Hill South Tower | 184.4 |
| 9 | 44 Montgomery | 172.3 | 101 California Street | 183 |
| 10 | Spear Tower | 172 | 50 Fremont Center | 183 |

*3.4. Oceana*

3.4.1. Melbourne

Melbourne, which is the coastal capital of the southeastern Australian state of Victoria, has been very active in constructing tall buildings. It has begun constructing the 317 m (1040 ft), 100-story Australia 108, which will be the tallest building in Oceana when completed in 2020 [6]. Other notable towers under construction include the 88-story Aurora Melbourne Central (2019), the 78-story Premier Tower (2020), the 75-story Victoria One (2018), and the 71-story Swanston Central (2019). Recently, Melbourne completed the 69-story Vision Apartments (2016), the 68-story 568 Collins Street (2015), and the 59-story Abode318 (2015). Melbourne will build more distinct skyscrapers including the Collins Arch and Premier Tower soon [2]. Overall, in the first two decades of the 21st century, Melbourne will add 116 skyscrapers to the 36 skyscrapers it built previously. The overwhelming majority of these skyscrapers are located in the city's downtown and along the Yarrar River. Fourteen of the 200 m+ are located near the upper and lower edges of the downtown including one supertall. Melbourne has currently 48 tall buildings under construction. Overall, this massive addition in the heart of

the downtown strengthens the image of Melbourne's city center (Figure 13). The average height of the 10 tallest skyscrapers in 2000 and 2020 will increase from 216 m to 266 m, equaling a 123% increase (Table 23).

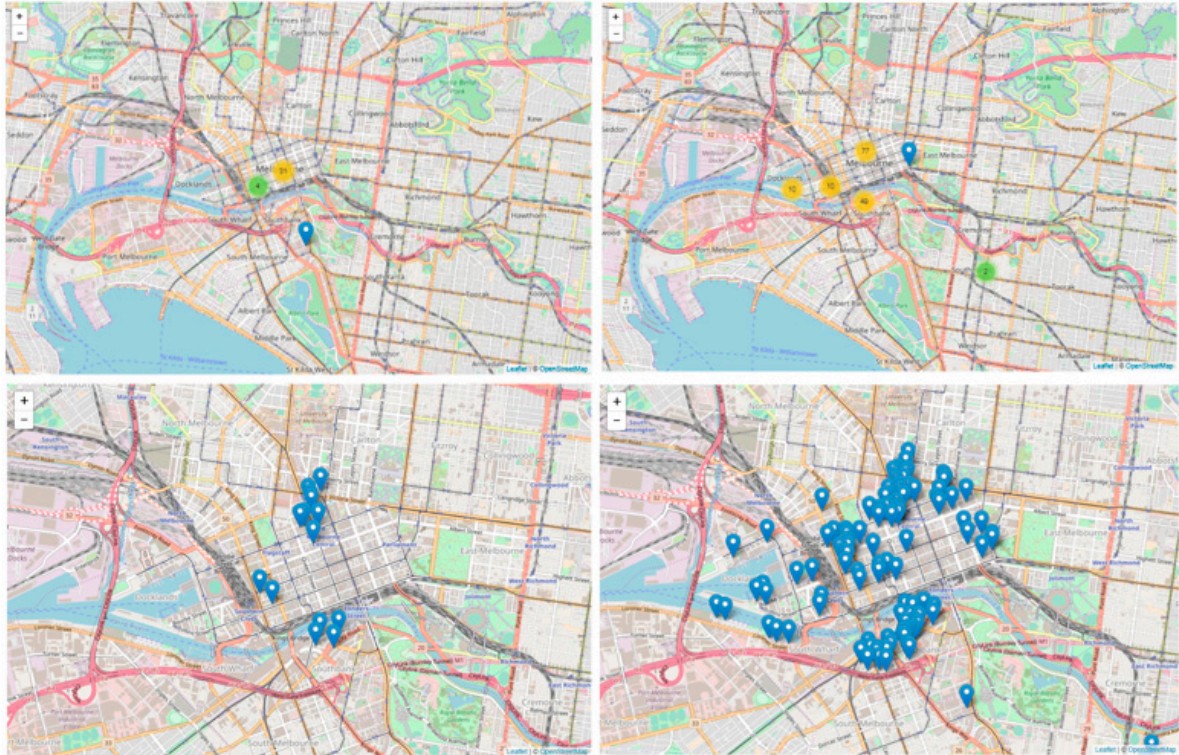

**Figure 13.** Melbourne in 2000 and 2020.

**Table 23.** Melbourne's 10 tallest skyscrapers in 2000 and 2020.

| # | Building Name | Height (m) | Building Name | Height (m) |
|---|---|---|---|---|
| 1 | 120 Collins Street | 264.9 | Australia 108 | 316.7 |
| 2 | 101 Collins Street | 260 | Eureka Tower | 297.3 |
| 3 | Rialto Towers | 251.1 | Aurora Melbourne Central | 269.6 |
| 4 | Bourke Place | 224 | 120 Collins Street | 264.9 |
| 5 | Telstra Corporate Building | 218 | 101 Collins Street | 260 |
| 6 | Melbourne Central | 211 | Prima Pearl Apartments | 254 |
| 7 | Sofitel Hotel—Collins Place | 188 | Rialto Towers | 251.1 |
| 8 | ANZ Tower—Collins Place | 188 | Queens Place North Tower | 249.9 |
| 9 | 80 Collins Street (Old Tower) | 182 | Premier Tower | 248.6 |
| 10 | 385 Bourke Street | 169 | Victoria One | 246.8 |

### 3.4.2. Sydney

Sydney continues to experience urban and economic growth. However, it is physically constrained by the harbor, which has reinforced building skyward. Sydney has started on constructing the 271 m (889 ft) Crown Sydney Hotel and Resort to become the tallest in the city and the second tallest in Oceana when completed in 2019. Other notable towers in Sydney under construction include the 68-story Sydney Greenland Center (2019), the 54-story 189 Macquarie Street North Tower (2018), and the 46-story Landmark (2018). In recent years, the city completed the 51-story International Towers Tower 1 (2016), the 43-story International Towers Tower 2 (2015), the 40-story International Towers Tower 3 (2016), and the 40-story 200 George Street (2016) [16,21]. Overall, in the first two decades of the 21st Century, Sydney will add 62 skyscrapers to the 83 skyscrapers it built previously. Forty of the new skyscrapers are located in the downtown near the older ones. Additionally, all nine 200 m+

are located in the same area as well. The rest of the added skyscrapers (22) are dispersed throughout the city. Sydney has no supertalls. Consequently, the added 40 skyscrapers in the downtown area strengthen its image as the city's center (Figure 14). The average height of the 10 tallest skyscrapers in 2000 and 2020 will increase from 208 m to 236 m, equaling a 114% increase (Table 24) [2,15].

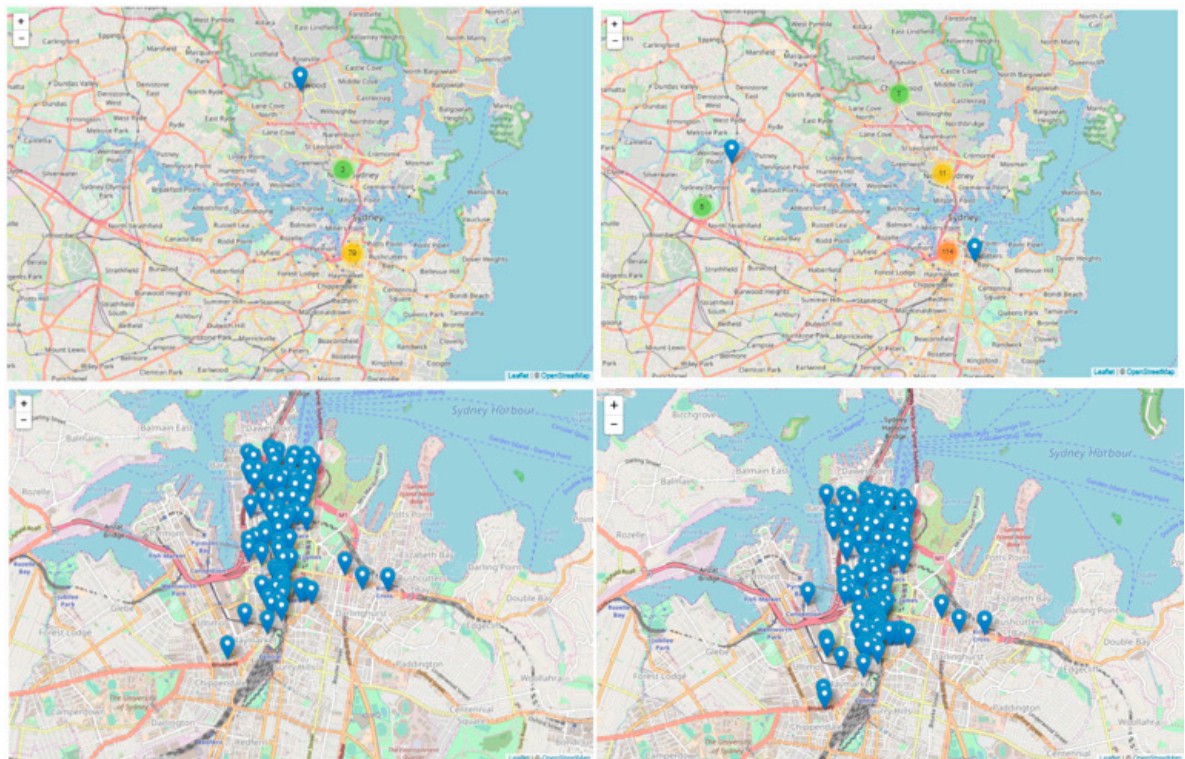

**Figure 14.** Sydney in 2000 and 2020.

**Table 24.** Sydney's 10 tallest skyscrapers in 2000 and 2020.

| # | Building Name | Height (m) | Building Name | Height (m) |
|---|---|---|---|---|
| 1 | Chifley Tower | 244 | Crown Sydney Hotel and Resort | 271.3 |
| 2 | Citigroup Center | 243 | Chifley Tower | 244 |
| 3 | MLC Center | 228 | Citigroup Centre | 243 |
| 4 | Governor Phillip Tower | 227 | Deutsche Bank Place | 240 |
| 5 | Aurora Place | 218.9 | Sydney Greenland Center | 236.5 |
| 6 | Suncorp Place | 193 | World Tower | 230 |
| 7 | AMP Center | 188 | MLC Center | 228 |
| 8 | Century Tower | 183 | Governor Phillip Tower | 227 |
| 9 | Grosvenor Place | 180 | Ernst & Young Tower at Latitude | 222 |
| 10 | Castlereagh Center | 173 | Aurora Place | 218.9 |

*3.5. The Middle East*

3.5.1. Dubai

Among all Middle Eastern cities, Dubai has been the most conspicuous "instant" skyscraper city. Not long ago, Dubai was an unnoticeable, small village with no tall buildings at all. Today, it is a home to the world's tallest building known as the 828 m (2716 ft) Burj Khalifa as well as other remarkable skyscrapers [6,20,21]. Today, Dubai is home to the greatest number of 200 m+ and 300 m+ buildings, which surpasses major skyscraper cities in the world including Hong Kong, New York City, Chicago, and Shanghai. Among the remarkable towers under construction are the 111-story Entisar Tower (2020), the 104-story Marina 106 (2019), the 80-story S Residence (2020), 74-story The Address Jumeirah

Resort and Spa at Jumeirah Beach Residence (2020), the 74-story Al Habtoor City Tower 1 (2017), the 74-story Al Habtoor City Tower 2 (2017), and the 20-story Opus (2018). Overall, in the first two decades of the 21st Century, Dubai will add 319 skyscrapers to the 28 skyscrapers it built previously. 144 of these skyscrapers are built along Shaikh Zayed road by stretching southwest and connecting to the new "downtown" near Burj Khalifa (Figure 15). Significantly, 174 skyscrapers were built away from the downtown (about 10 miles southwest) in new districts such as Dubai Marina [2]. Fifty-seven 200 m+ skyscrapers (including 17 supertalls) are placed near Burj Khalifa while 34 skyscrapers (including 10 supertalls) are located in Dubai Marina. The average height of the 10 tallest skyscrapers in 2000 and 2020 will increase from 198 m to 424 m, equaling a 213% increase (Table 25).

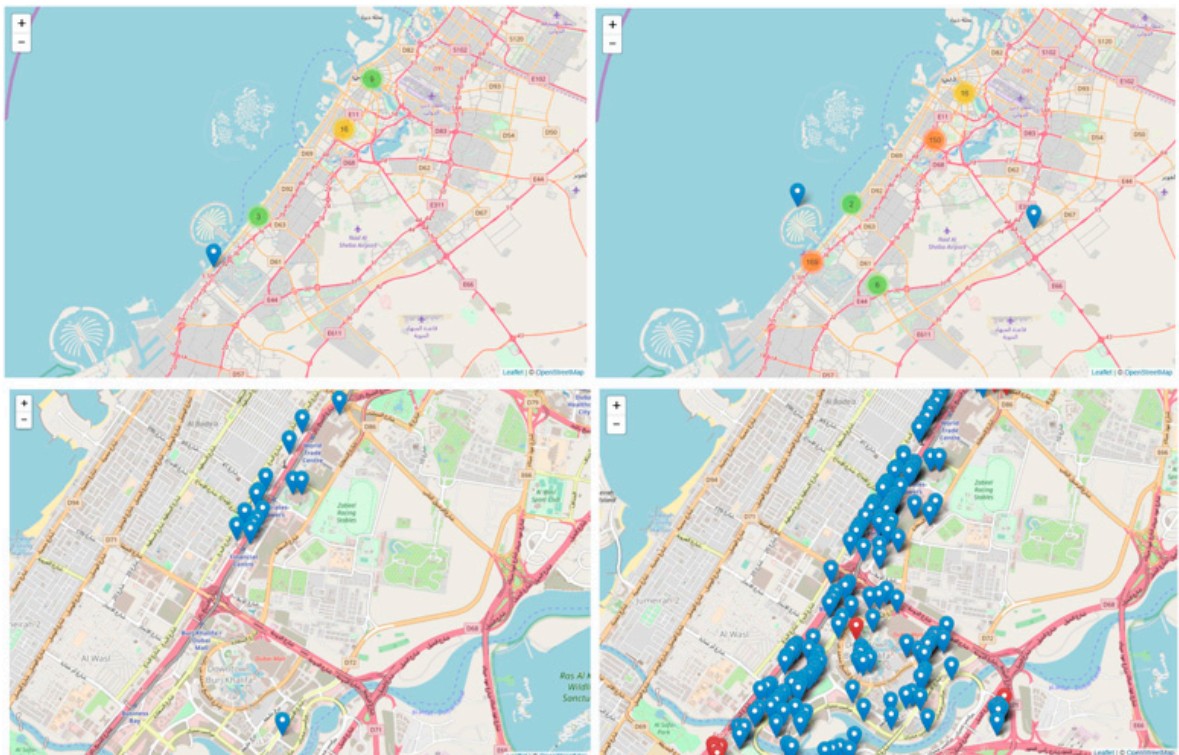

**Figure 15.** Dubai in 2000 and 2020.

**Table 25.** Dubai's 10 tallest skyscrapers in 2000 and 2020.

| # | Building Name | Height (m) | Building Name | Height (m) |
|---|---|---|---|---|
| 1 | Emirates Tower One | 354.6 | Burj Khalifa | 828 |
| 2 | Burj Al Arab | 321 | Marina 101 | 425 |
| 3 | Emirates Tower Two | 309 | Princess Tower | 413.4 |
| 4 | Al Saqr Business Tower | 158 | 23 Marina | 392.4 |
| 5 | Saeed Tower I | 150 | Elite Residence | 380.5 |
| 6 | World Trade Center | 149 | The Address Boulevard | 370 |
| 7 | Paloma Tower | 146.3 | Almas Tower | 360 |
| 8 | Deira Tower | 135 | Gevora Hotel | 356.3 |
| 9 | Ghaya Residence | 131 | JW Marriott Marquis Hotel Dubai Tower 1 | 355.4 |
| 10 | API World Tower | 130.5 | JW Marriott Marquis Hotel Dubai Tower 2 | 355.4 |

### 3.5.2. Doha

Doha, the capital of Qatar, is also among the most active cities in building significant skyscrapers in the Middle East. In the first two decades of the 21st Century, Doha will add 48 skyscrapers to the only one it built previously. Overall, all these new skyscrapers concentrate in the downtown region

(Figure 16). The average height of the 10 tallest skyscrapers in 2000 and 2020 will increase from 115 m to 243 m, equaling a 211% increase (Table 26).

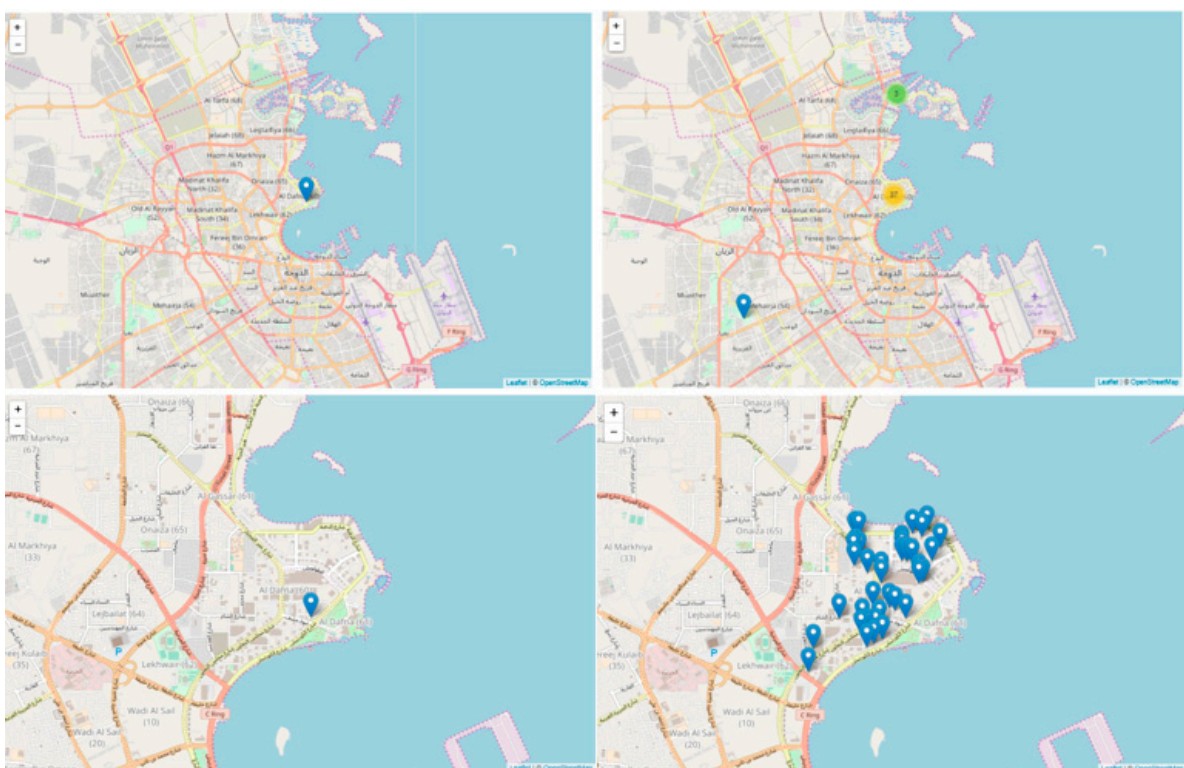

**Figure 16.** Doha in 2000 and 2020.

**Table 26.** Doha's 10 tallest skyscrapers in 2000 and 2020.

| # | Building Name | Height (m) | Building Name | Height (m) |
|---|---|---|---|---|
| 1 | Ritz Carlton Hotel | 115 | Aspire Tower | 300 |
| 2 | N.A. | N.A. | Kempinski Residences and Suites | 253.3 |
| 3 | N.A. | N.A. | Palm Tower 1 | 244.5 |
| 4 | N.A. | N.A. | Palm Tower 2 | 244.5 |
| 5 | N.A. | N.A. | World Trade Center Doha | 241.1 |
| 6 | N.A. | N.A. | Doha Tower | 238.1 |
| 7 | N.A. | N.A. | Hilton Double Tree Sinyar Tower | 230 |
| 8 | N.A. | N.A. | Al Faisal Tower | 227 |
| 9 | N.A. | N.A. | Al Asmakh Tower | 227 |
| 10 | N.A. | N.A. | Qatar Petroleum District Tower 7 | 222.8 |

### 3.5.3. Riyadh

Riyadh, the Capital of Saudi Arabia, is also experiencing an unprecedented pace of building significant skyscrapers. Being the seat of government, this "conservative" city has recently broken its long tradition of maintaining a low-rise city profile. In the first two decades of the 21st Century, Riyadh will add 43 buildings to the only three skyscrapers it built previously. Interestingly, all new skyscrapers attain a linear stretch along a major downtown boulevard (Figure 17). The average height of the 10 tallest skyscrapers in 2000 and 2020 will increase from 156 m to 278 m, equaling a 178% increase (Table 27).

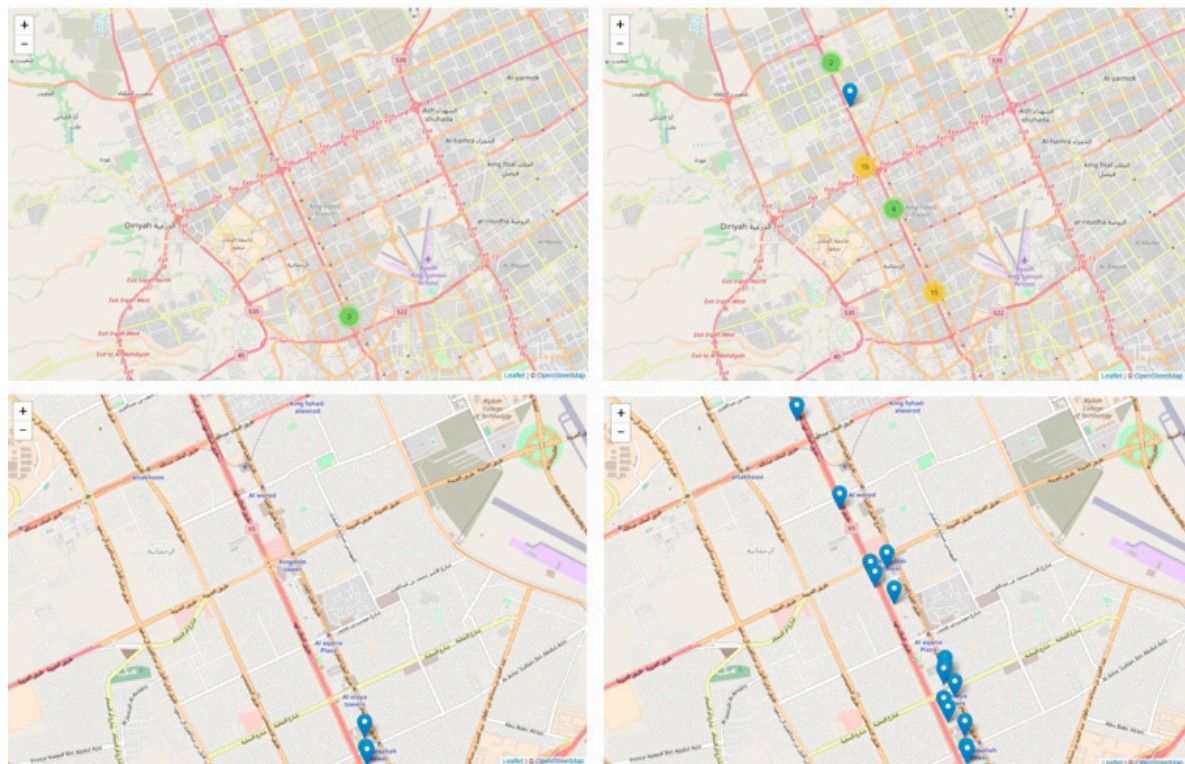

**Figure 17.** Riyadh in 2000 and 2020.

**Table 27.** Riyadh's 10 tallest skyscrapers in 2000 and 2020.

| # | Building Name | Height (m) | Building Name | Height (m) |
|---|---|---|---|---|
| 1 | Al Faisaliah Center | 266.9 | PIF Tower | 385 |
| 2 | NCCI Towers North | 100.6 | Burj Rafal | 307.9 |
| 3 | NCCI Towers South | 100.6 | KAFD World Trade Center | 304 |
| 4 | N.A. | N.A. | Kingdom Centre | 302.3 |
| 5 | N.A. | N.A. | Al Faisaliah Center | 266.9 |
| 6 | N.A. | N.A. | GCC Bank Headquarters | 264 |
| 7 | N.A. | N.A. | Tamkeen Tower | 258.2 |
| 8 | N.A. | N.A. | Al Majdoul Tower | 244 |
| 9 | N.A. | N.A. | Samba Bank HQ Tower | 231.2 |
| 10 | N.A. | N.A. | Rafal Living Tower | 213 |

### 3.5.4. Tel Aviv

As Tel Aviv urbanizes rapidly, its skyline exhibits more tall buildings. Among the recently topped out towers include Azrieli Sarona (238 m (781 ft)), the Midtown Office Tower (196 m (643 ft)), and the Alon Tower A & B (162 m (530 ft)). Towers under construction include the Midtown Residential Tower (196 m (643 ft)) and the Sitonai Market Tower I & II (160 m (525 ft)). Overall, in the first two decades of the 21st Century, Tel Aviv will add 42 skyscrapers to the 11 skyscrapers it built previously. By then, the city will have only two 200 m+ skyscrapers called the Azrieli Sarona (2017) and ToHA Tower 1 (2020) [2]. The new skyscrapers follow a linear stretch (Figure 18). The average height of the 10 tallest skyscrapers in 2000 and 2020 will increase from 136 m to 192 m, equaling a 142% increase (Table 28).

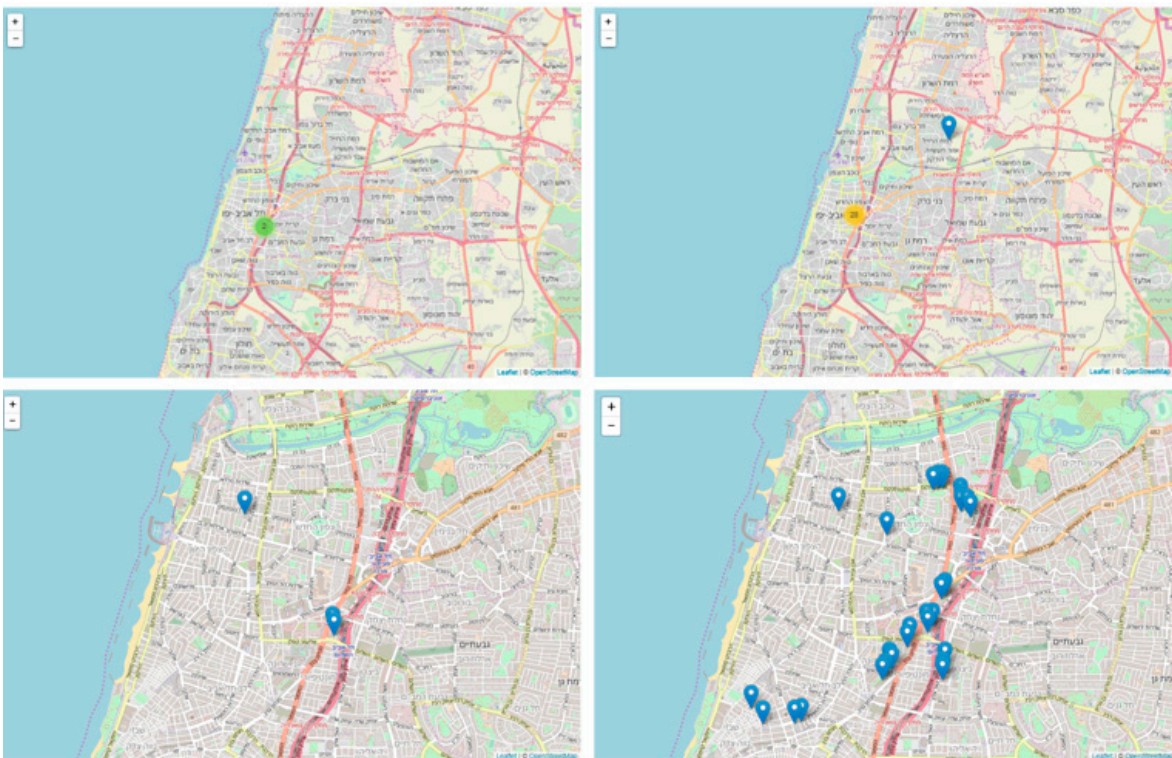

**Figure 18.** Tel Aviv in 2000 and 2020.

**Table 28.** Tel Aviv's 10 tallest skyscrapers in 2000 and 2020.

| # | Building Name | Height (m) | Building Name | Height (m) |
|---|---|---|---|---|
| 1 | Azrieli Center Circular Tower | 187 | ToHA Tower 1 | 285 |
| 2 | Azrieli Center Triangular Tower | 169 | Azrieli Sarona | 238.4 |
| 3 | Shalom Meir Tower | 142 | Midtown Office Tower | 196 |
| 4 | Tel Aviv Towers I | 140 | Midtown Residential Tower | 196 |
| 5 | Tel Aviv Towers II | 140 | Azrieli Center Circular Tower | 187 |
| 6 | Marganit Tower | 138 | Azrieli Center Triangular Tower | 169 |
| 7 | Levinstein Tower | 124 | Electra Tower | 169 |
| 8 | Isrotel Tower | 108 | Alon Tower A | 161.7 |
| 9 | Dizengoff Tower | 106 | Alon Tower B | 161.7 |
| 10 | Rubenstein House | 102 | Sitonai Market Tower I & II | 160 |

## 3.6. South America

São Paulo

Overall, in the first two decades of the 21st Century, São Paulo, Brazil, will add 70 skyscrapers to the 17 skyscrapers it built previously. While some of the new skyscrapers reinforce existing "downtown" clusters, others have created a new significant linear stretch (Figure 19). The average height of the 10 tallest skyscrapers in 2000 and 2020 will increase from 148 m to 160 m, equaling a 108% increase (Table 29).

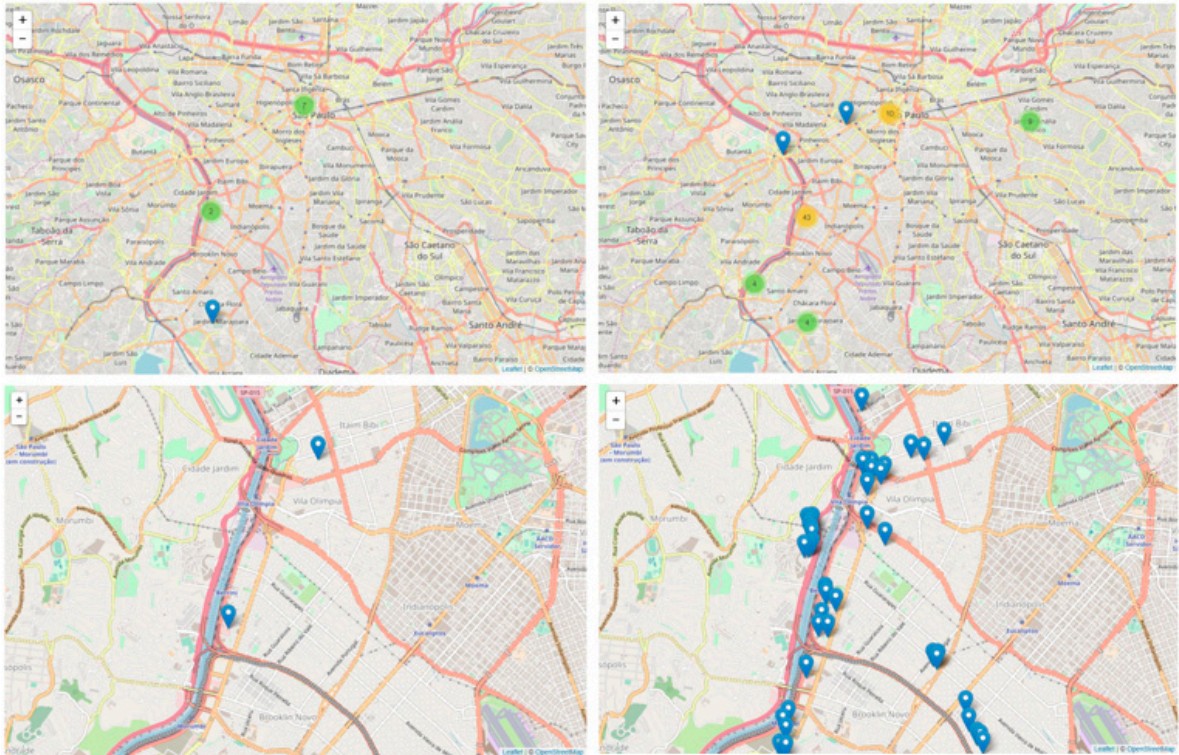

**Figure 19.** São Paulo in 2000 and 2020.

**Table 29.** São Paulo's 10 tallest skyscrapers in 2000 and 2020.

| # | Building Name | Height (m) | Building Name | Height (m) |
|---|---|---|---|---|
| 1 | Palácio W. Zarzur | 170 | Palácio W. Zarzur | 170 |
| 2 | Edificio Italia | 165 | Edificio Italia | 165 |
| 3 | Altino Arantes | 161 | Altino Arantes | 161 |
| 4 | Torre Norte | 158 | Torre Norte | 158 |
| 5 | Birmann 21 | 149 | Begonias | 157.9 |
| 6 | Banco do Brasil | 143 | Ipes | 157.9 |
| 7 | Plaza Centenario | 139 | Jabuticabeiras | 157.9 |
| 8 | Barao de Iguape | 133 | Limantos | 157.9 |
| 9 | Ipiranga 165 | 130 | Magnolias | 157.9 |
| 10 | Grande São Paulo | 129 | Reseda | 157.9 |

## 3.7. Central America

### 3.7.1. Panama City

Panama City, which is the capital and largest city of Panama, experiences a rapid pace in building tall buildings. Among the recently completed towers in Panama City are: Trump Ocean Club International Hotel & Tower (2011), Torre Vitri (2012), Bicsa Financial Center (2013), and The Point (2013) [2]. Overall, in the first two decades of the 21st Century, Panama City will add 63 skyscrapers to the seven skyscrapers it built previously. While some of the new skyscrapers reinforce the existing center, others form a new one, thereby initiating a polycentric city (Figure 20). The average height of the 10 tallest skyscrapers in 2000 and 2020 will increase from 156 m to 253 m, equaling a 162% increase (Table 30).

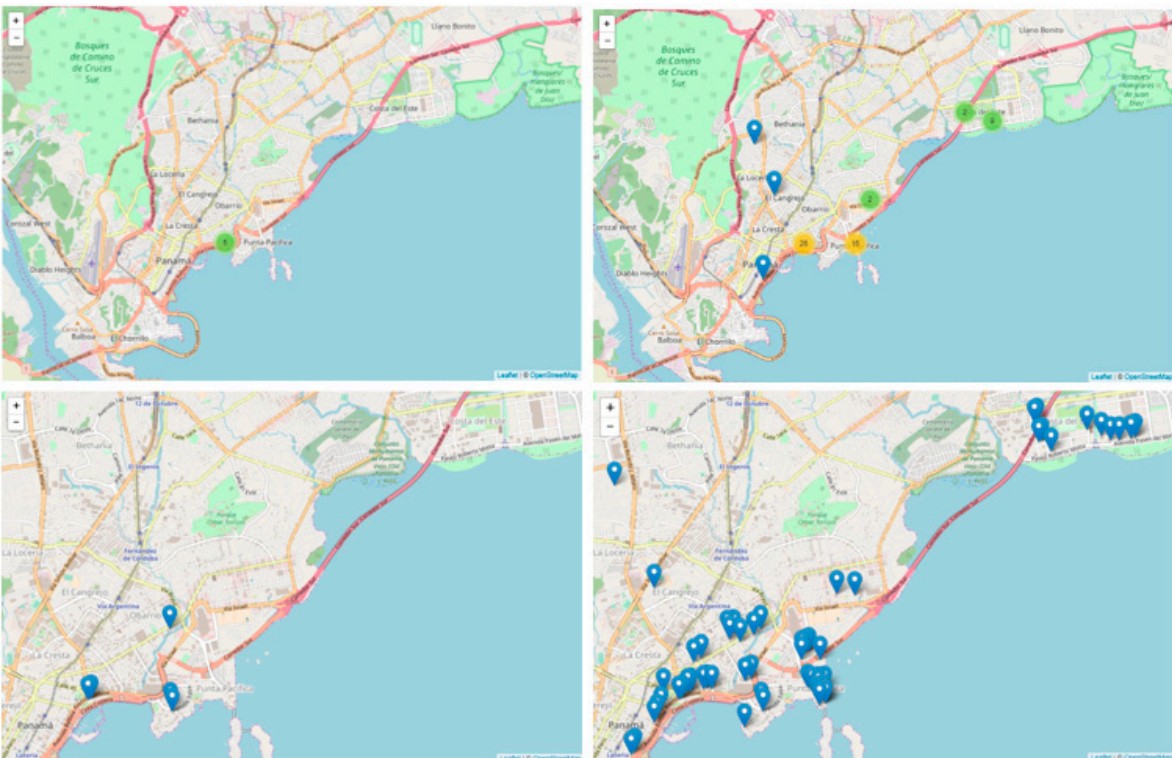

**Figure 20.** Panama City in 2000 and 2020.

**Table 30.** Panama City's 10 tallest skyscrapers in 2000 and 2020.

| # | Building Name | Height (m) | Building Name | Height (m) |
|---|---|---|---|---|
| 1 | Torre Mirage | 172 | JW Marriott | 284 |
| 2 | Miramar Towers I | 168 | Torre Vitri | 280.7 |
| 3 | Miramar Towers II | 168 | Bicsa Financial Center | 267 |
| 4 | Platinum Tower | 158.5 | The Point | 266 |
| 5 | PH Plaza Credicorp Bank Panama | 146 | YooPanama Inspired by Starck | 246.8 |
| 6 | Coastal Tower | 144 | Ocean Two | 245.7 |
| 7 | HSBC Tower | 135 | Pearl Tower | 242.2 |
| 8 | N.A. | N.A. | Rivage | 233.2 |
| 9 | N.A. | N.A. | F&F Tower | 232.7 |
| 10 | N.A. | N.A. | Torre Waters | 232 |

### 3.7.2. Mexico City

Mexico City, which is the capital of Mexico and the most populous city in North America, also experienced a rapid pace of building tall towers. In the first two decades of the 21st Century, Mexico City will add 53 skyscrapers to the 27 skyscrapers that it built previously. Most of the new skyscrapers reinforce existing clusters (Figure 21). The average height of the 10 tallest skyscrapers in 2000 and 2020 will increase from 160 m to 213 m, equaling a 133% increase (Table 31).

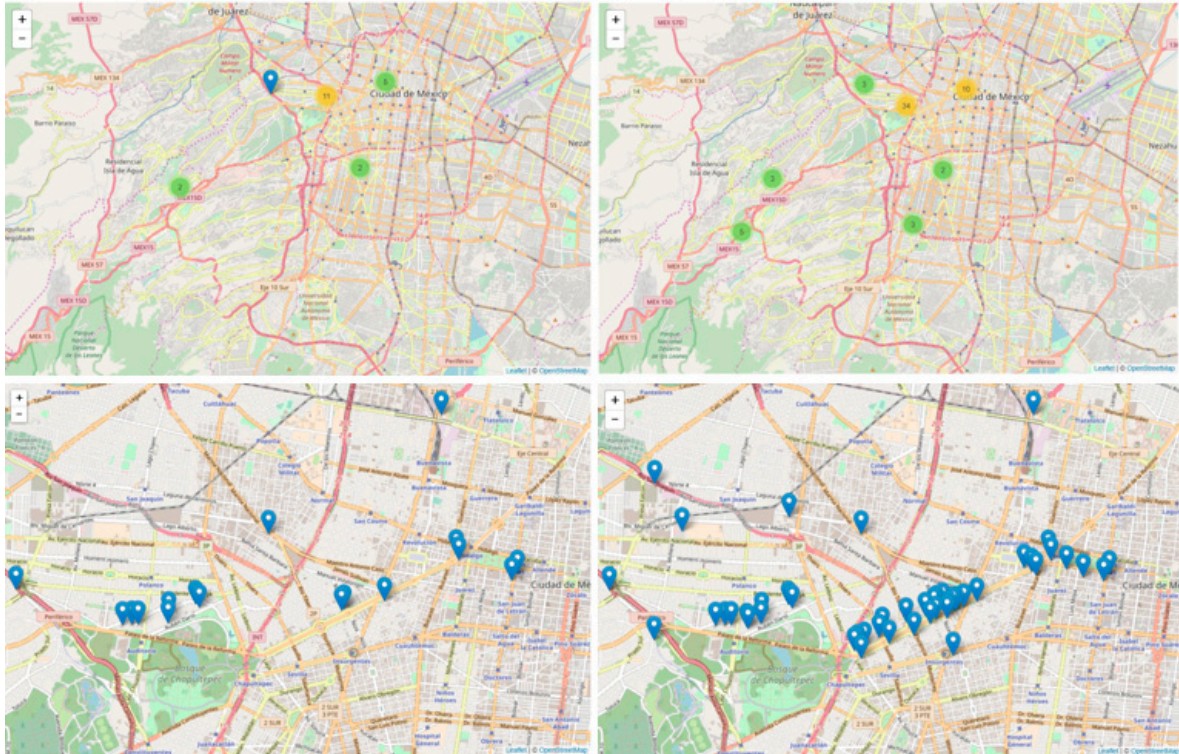

**Figure 21.** Mexico City in 2000 and 2020.

**Table 31.** Mexico City's 10 tallest skyscrapers in 2000 and 2020.

| # | Building Name | Height (m) | Building Name | Height (m) |
|---|---|---|---|---|
| 1 | Torre Ejecutiva Pemex | 211.3 | Torre Reforma | 246 |
| 2 | Torre Altus | 195 | Chapultepec Uno | 241 |
| 3 | Torre Latinoamerica | 182 | Torre BBVA Bancomer | 234.9 |
| 4 | World Trade Center | 172.3 | Torre Paradox | 234 |
| 5 | Arcos Torre I | 161.2 | Torre Mayor | 225 |
| 6 | Torre Lomas | 146.5 | Torre Ejecutiva Pemex | 211.3 |
| 7 | Hyatt Regency Mexico City | 136 | Torre Altus | 195 |
| 8 | Torre Del Caballito | 135 | Torre Reforma Latino | 185 |
| 9 | Torre Mural | 133 | Torre Latinoamerica | 182 |
| 10 | Torre Mexicano de Aviacion | 132 | Torres Cuarzo Office Tower | 180 |

*3.8. Africa*

Nairobi

Nairobi, which is the capital and the largest city of Kenya, experienced a moderate pace of building skyscrapers. In the first two decades of the 21st century, Nairobi will add six skyscrapers to the four skyscrapers it built previously. Most of these skyscrapers reinforce existing locations of tall buildings, which improve the image of the downtown (Figure 22). The average height of the 10 tallest skyscrapers in 2000 and 2020 will increase from 117 m to 159 m, equaling a 159% increase (Table 32).

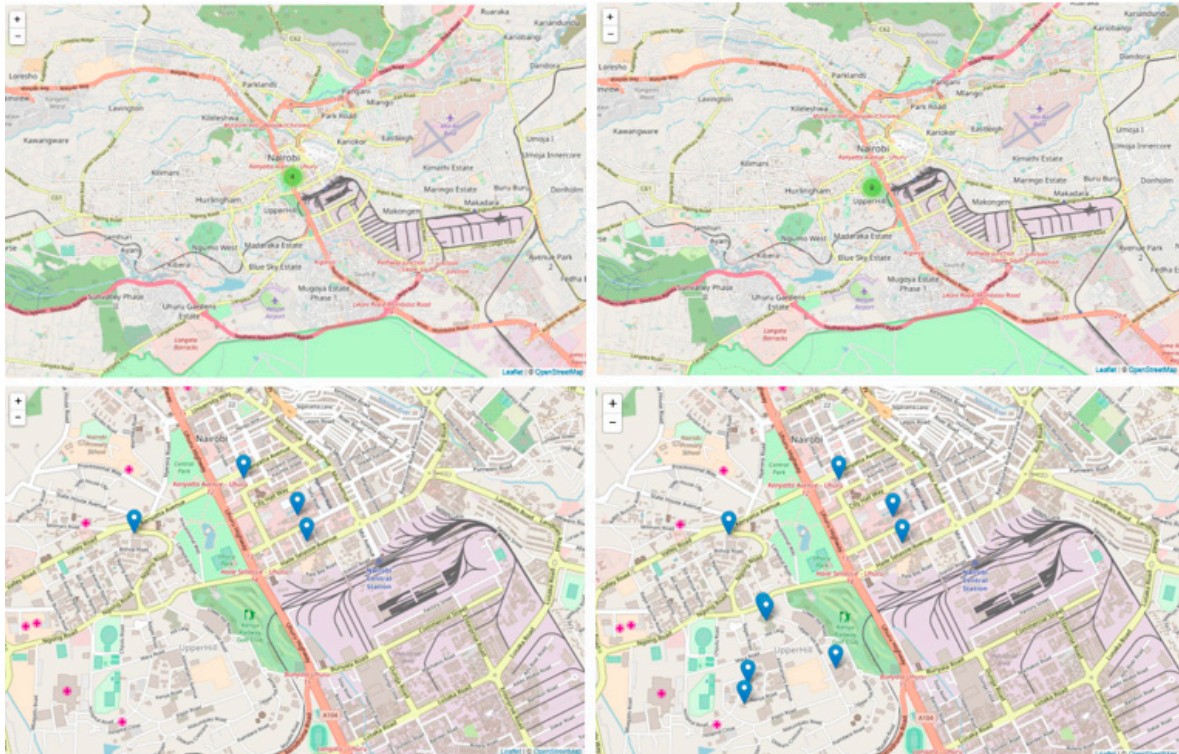

**Figure 22.** Nairobi in 2000 and 2020.

**Table 32.** Nairobi's 10 tallest skyscrapers in 2000 and 2020.

| # | Building Name | Height (m) | Building Name | Height (m) |
|---|---|---|---|---|
| 1 | Times Tower | 140 | The Pinnacle Tower | 320 |
| 2 | Teleposta Towers | 120 | Pinnacle Tower II | 201 |
| 3 | Kenyatta International Conference Center | 105.2 | Britam Tower | 200.1 |
| 4 | Social Security House | 102.7 | UAP Old Mutual Tower | 163 |
| 5 | N.A. | N.A. | Times Tower | 140 |
| 6 | N.A. | N.A. | Parliament Tower | 125 |
| 7 | N.A. | N.A. | Teleposta Towers | 120 |
| 8 | N.A. | N.A. | Kenya Commercial Bank Plaza | 109.1 |
| 9 | N.A. | N.A. | Kenyatta International Conference Center | 105.2 |
| 10 | N.A. | N.A. | Social Security House | 102.7 |

## 4. Discussion

### 4.1. Counts

Upon reviewing the examined cities, we find that in the 21st Century all cities have been adding significant numbers of skyscrapers. Cities that added the greatest number of skyscrapers follow this order: Dubai, New York City, Shenzhen, Shanghai, Moscow, Chicago, Melbourne, Bangkok, Miami, London, São Paulo, Panama City, Sydney, Beijing, Mexico City, Doha, Riyadh, Tel Aviv, San Francisco, and Nairobi (Table 33). When we calculate the percentage of increase of skyscrapers, we find that:

- seven cities (Doha, Riyadh, Dubai, Panama City, Shenzhen, and Beijing) have experienced a 550%–4800% increase,
- eight cities (Moscow, São Paulo, Tel Aviv, Bangkok, London, Miami, Melbourne, and Shanghai) have experienced a 226%–487% increase, and
- six cities (Mexico City, Nairobi, Sydney, Chicago, New York City, and San Francisco) have experienced a 43%–196% increase.

These findings confirm the views that the Chinese and Middle Eastern cities are the "truest" emerging skyscraper cities in the 21st century. The percentages of increase (550%–4800%) are very remarkable to the extent that one may question the soundness of these activities. The urban transition is so quick and immense and, therefore, some unintended consequences are inevitable. For example, recent research indicates that millions of homes in new high-rise buildings in China are vacant, which highlights the problem of building too many high-rises in a very short period of time [46].

Certainly, the Chinese government has enormously promoted constructing high-rise cities to house new massive urban population flocking from rural areas. However, Chinese people have largely shunned these developments because they disliked the design, layout, architectural styles, schools, and amenities. Certainly, these cities were developed in a rushed manner, attempting to house hundreds of thousands of people in a few years and, consequently, city planners disengaged residents from the design process and deprived them from voicing their preferences. The design process did not take into consideration that many of the intended inhabitants were villagers who were accustomed to low-rise living instead of high-rise living [46].

Ordos Kangbashi in China is one of several new high-rise cities sitting almost vacant because of these reasons. Although Ordos Kangbashi offers beautiful architecture, attractive plazas, and state-of the-art recreational spaces, this stillborn city lacks the essential ingredients of success including affordability, socio-economic vitality, and employment opportunities [46]. Furthermore, Ordos Kangbashi's location is detrimental for being remote from any settlements. It is located in the Inner Mongolia's vast steppe that features a harsh climate and lacks basic life essentials such as water and vegetation. Consequently, while planners have designed this new city to house over a million of inhabitants, barely a few thousand people ever settled.

Chinese people have nicknamed these cities "modern ghost cities" because they evoke an eerie sensation promoted by silent streets, vacant high-rises, empty parks, and dead public spaces. In turn, this prevailing negative image has further discouraged people from considering moving into these new high-rise cities [46]. This local phenomenon has promoted global consciousness and a negative image about high-rise developments.

**Table 33.** Skyscrapers in 2000 and 2020.

| # | City | Number of Skyscrapers in 2000 | Number of Skyscrapers Built between 2000 and 2020 | Number of Skyscrapers in 2020 | Percentage of Increase |
|---|---|---|---|---|---|
| 1 | Dubai | 25 | 322 | 347 | 1288 |
| 2 | New York City | 590 | 305 | 895 | 52 |
| 3 | Shenzhen | 20 | 173 | 193 | 865 |
| 4 | Shanghai | 69 | 157 | 226 | 228 |
| 5 | Moscow | 31 | 151 | 182 | 487 |
| 6 | Chicago | 207 | 127 | 334 | 61 |
| 7 | Melbourne | 36 | 116 | 152 | 322 |
| 8 | Bangkok | 27 | 96 | 123 | 356 |
| 9 | Miami | 27 | 92 | 119 | 341 |
| 10 | London | 23 | 80 | 103 | 348 |
| 11 | São Paulo | 17 | 70 | 87 | 412 |
| 12 | Panama City | 7 | 63 | 70 | 900 |
| 13 | Sydney | 83 | 61 | 144 | 74 |
| 14 | Beijing | 10 | 60 | 70 | 600 |
| 15 | Mexico City | 27 | 53 | 80 | 196 |
| 16 | Doha | 1 | 48 | 49 | 4800 |
| 17 | Riyadh | 3 | 43 | 46 | 1433 |
| 18 | Tel Aviv | 11 | 42 | 53 | 382 |
| 19 | San Francisco | 67 | 29 | 96 | 43 |
| 20 | Nairobi | 4 | 6 | 10 | 150 |

*4.2. Heights*

Similarly, upon reviewing skyscraper developments in the examined cities, we find that all of them have experienced a significant height increase (Table 34). Interestingly, the increased height,

as has been critiqued in literature [2], is largely unnecessary. Many of the tallest buildings in these cities suffer from a "vanity height" problem. The CTBUH has coined the "vanity height" term to refer to the wasted space between a skyscraper's highest occupiable floor and its architectural top [47]. For example, the average vanity ratio in the UAE is 19%, which makes it the nation with the "vainest" tall buildings. The Burj Khalifa's vanity height is 244 m (800 ft), which qualifies to be a skyscraper on its own. Burj Al Arab in Dubai, UAE, has a 39% ratio (124 m:321 m), (407 ft:1053 ft)—the greatest "vanity ratio" among completed supertalls [47].

**Table 34.** Increased height of the 10 tallest skyscrapers in 2000 and 2020.

| # | City | 2000 Average (m) | 2020 Average (m) | Height Increase (%) |
|---|---|---|---|---|
| 1 | Dubai | 198 | 424 | 213 |
| 2 | Doha | 115 | 243 | 211 |
| 3 | Beijing | 147 | 291 | 200 |
| 4 | London | 142 | 281 | 198 |
| 5 | Moscow | 164 | 314 | 191 |
| 6 | Riyadh | 156 | 278 | 178 |
| 7 | Shenzhen | 231 | 393 | 170 |
| 8 | Panama City | 156 | 253 | 162 |
| 9 | Shanghai | 235 | 365 | 160 |
| 10 | Nairobi | 117 | 159 | 159 |
| 11 | Miami | 154 | 219 | 142 |
| 12 | Tel Aviv | 136 | 192 | 142 |
| 13 | Bangkok | 205 | 274 | 134 |
| 14 | Mexico City | 160 | 213 | 133 |
| 15 | New York | 314 | 398 | 127 |
| 16 | Melbourne | 216 | 266 | 123 |
| 17 | San Francisco | 194 | 221 | 114 |
| 18 | Sydney | 208 | 236 | 114 |
| 19 | São Paulo | 148 | 160 | 108 |
| 20 | Chicago | 308 | 326 | 106 |

*4.3. Mixed-Use Towers*

Upon reviewing the changing functional use of skyscrapers in three height categories (100 m+, 200 m+, and 300 m+) in 2000 and 2020, we find that for the 100 m+ skyscrapers, the office space portion has been reduced from 71% to 32% while residential space has been increased from 13% to 44%. In addition, mixed-use space has been increased from 9% to 19% while hotel space remained unchanged. Similarly, 200 m+ skyscrapers experienced a drop in office space from 82% to 40% while residential space has increased from 3% to 28%. Mixed-use space has also increased from 12% to 27%. Hotel space remained unchanged. Lastly, 300 m+ skyscrapers experienced a drop in office space from 75% to 32% while residential space increases from 0% to 13%. Significantly, mixed-use space has increased from 12% to 50% while hotel space has been dropped from 13% to 5% (Table 35).

Therefore, these findings confirm the view that mixed-use towers have become a prevailing trend in the 21st century. A mixed-use tower could be more sustainable than a single-use tower for multiple reasons such as economic uncertainty and a fluctuating market. Commercial synergy results from mixed functions, adaptive reuse, convenience, and smaller plates at upper floors. In an unstable economy, a mixed-use building offers greater opportunities to secure investment in real estate development because the rental income comes from multiple sources. Second, multiple uses secure the presence of people and economic activities for longer hours—potentially around the clock. This provides convenience to local tenants and improves perceived safety and security.

**Table 35.** Changing functional use of skyscrapers in 2000 and 2020.

| Function | 100 m+ Skyscrapers | | 200 m+ Skyscrapers | | 300 m+ Skyscrapers | |
|---|---|---|---|---|---|---|
| Year | 2000 | 2020 | 2000 | 2020 | 2000 | 2020 |
| Office | 71% | 32% | 82% | 40% | 75% | 32% |
| Residential | 13% | 44% | 3% | 28% | 0% | 13% |
| Mixed use | 9% | 19% | 12% | 27% | 12% | 50% |
| Hotel | 4% | 4% | 3% | 3% | 13% | 5% |
| Others | 3% | 1% | 0% | 1% | 0% | 0% |

David Nicholson-Cole concisely captures these notions in an article *Rise of the Glass Giants: How Modern Cities are Forcing Skyscrapers to Evolve*. He explains: "Mixed-use towers make the best use of land and are more resilient to economic shocks because the rental income comes from lots of different sources—and the flows of people are balanced instead of peaking twice daily" [48]. Furthermore, mixed-use towers have the potential to use resources and waste efficiently. For example, the water system can capture graywater from residential spaces (which generate a larger amount of graywater) and transfer recovered water to the cooling system of office spaces where water consumption is high and potable water use is low. This type of system drastically reduces the use of potable water in office spaces (which is generally used for the cooling system); consequently, it results in significant savings.

Furthermore, as cities build taller buildings, the smaller floor plates at the upper floors fit functional use that requires fewer tenants such as in hotel rooms and small apartments. Smaller floor plates at upper floors are the results of structural requirements to withstand powerful wind at higher altitudes. In addition to residential and hotel spaces, upper floors can host amenities and services such as restaurants, café, sky gardens, and observation decks. For instance, The Shard in London integrates a viewing gallery at the upper floors. The Marina Bay Sands in Singapore incorporates a SkyPark that offers a wide range of amenities and the Wilshire Grand Center in Los Angeles, CA (completed recently) features a sky-deck with café and al fresco dining. Remarkably, Lotte World Tower dedicates its top 10 floors for extensive public use and entertainment facilities. Observation decks are a lucrative business and increasingly are coupled with amenities such as bars, restaurants, café, and interactive media displays, glass floors, and the like. "These unique spaces quench humanity's innate desire to see what the world looks like from the tops of these buildings and they, in turn, give viewers a new perspective on our cities" [49] (p. 13).

As such, over the past two decades, the mixed-use tower concept has been further developed and refined. Additional examples of mixed-use towers include Shanghai Tower, Singapore's PS100, Hong Kong's Hysan Place, Shenzhen's Hanking Center Tower (under construction), and the proposed redevelopment of the Khalil Gibran International Academy in New York City. Overall, the mixed-use tower represents a new architectural/planning paradigm that aims to create a sustainable "vertical neighborhood" or "vertical village" characterized by being self-contained, self-sufficient, and resilient [50].

*4.4. Residential Supertalls*

When we review functional change against height, we discover an important trend of building residential supertalls. In the past century, the tallest residential towers prevailed in pioneering vertical cities such as Hong Kong, New York City, and Chicago. Today, however, new places such as Dubai, Jeddah, and Abu Dhabi (Middle East), Shanghai and Beijing (China), and Mumbai (India) have joined the race of building the highest residential towers. Other cities that are building significant residential towers include Busan (South Korea), Moscow (Russia), Sydney, Gold Coast, and Melbourne (Australia), Bangkok (Thailand), Shanghai and Shenzhen (China), Istanbul (Turkey), and Toronto (Canada) among others [2,15].

This phenomenon started around the turn of the new century, but it slowed down after the 9/11 catastrophic event and the economic recession between 2007 and 2011. Later, taller residential

developments have picked up remarkable speed. Notably, many of the tallest buildings offer luxury apartments and condominiums [5,15]. Dubai and New York City have been among the most active cities in building residential supertalls. For example, Dubai has built the "world's tallest block" that contains some of the tallest residential towers including:

- Princess Tower (413 m/1356 ft)
- 23 Marina (393 m/1289 ft)
- Elite Residence (381 m/1250 ft)

  Similarly, New York City (NYC) has built remarkable residential towers in recent years, including:

- 423 Park Avenue (426 m/1396 ft)
- Eight Spruce Street (272 m/891 ft)
- 50 West (237 m/778 ft)

  Additional significant residential towers under construction in NYC include:

- 111 West 57th Street (435 m/1428 ft)
- 53 West 53rd (320 m/1050 ft)
- 220 Central Park South (290 m/950 ft)
- 15 Hudson Yards (279 m/914 ft)
- 125 Greenwich Street (274 m/898 ft)

## 4.5. Spatial Patterns

Upon reviewing the spatial patterns of skyscrapers, we find that, before the turn of the century, many cities featured fragmented arrangement of skyscrapers. Examples of these cities include Nairobi, Tel Aviv, Beijing, Bangkok, Doha, Panama City, São Paulo, Riyadh, London, Dubai, and Shenzhen. However, in 2020, the spatial distributions of skyscrapers in these cities vary since they became semi monocentric, monocentric, super monocentric, semi polycentric, polycentric, and super polycentric. Other cities change from monocentric to super monocentric, e.g., Miami, San Francisco, Sydney, and Melbourne while some cities such as Shanghai, Moscow, Chicago, and New York change from polycentric to super polycentric (Table 36).

However, a common theme among these various spatial patterns stresses the principle of clustering. As stated earlier, clustering of tall buildings fosters urban synergy among the provided diverse activities and specialized services. The high concentration of activities creates "knowledge spillovers" between firms in the same sector and across sectors that lead to increased innovation. Clustering also enhances the image of an existing downtown area or a new one. A group of tall buildings accentuates their height effect. In other words, the idea that the whole can be greater than the sum of the parts applies here. The great image of a modern downtown often comes from the cumulative total of its tall buildings fitting together instead of just from scattered buildings. Ideally, every tall building should be a new masterpiece of the evolving metropolis, enriching the ever-changing urban collage. This collective work of art should form an unflinching record of our scientific and artistic development. Many of the newly constructed tall buildings offer a fresh, modern style made possible by advancements in computer design and improvements in building materials.

Furthermore, the agglomeration of tall, mixed-use buildings in conjunction with mass-transit systems can provide a defined spatial structure for the city, which improves its urban functionality, boosts its commercial synergy, and enhances neighborhood place making. When we cluster tall buildings near mass-transit nodes, these buildings visually define the locations of these transit centers, which highlights their functional significance. In this way, taller buildings can provide reference points that make navigating the city more intuitive while drawing attention to mass-transit nodes. Because of their prominent height and physical distinction, clustered nodes create contrasts with low-rise

buildings and streetscapes. These nodes have the potential to break the visual monotony established by buildings of a similar height and architectural design [51].

**Table 36.** Changing spatial patterns of skyscrapers in 2000 and 2020.

| # | City | 2000 | 2020 |
|---|------|------|------|
| 1 | Shanghai | Polycentric | Super Polycentric |
| 2 | Beijing | Fragmented | Monocentric |
| 3 | Shenzhen | Fragmented | Super Polycentric |
| 4 | Bangkok | Fragmented | Monocentric |
| 5 | London | Fragmented | Polycentric |
| 6 | Moscow | Polycentric | Super Polycentric |
| 7 | New York | Polycentric | Super Polycentric |
| 8 | Chicago | Monocentric | Super Monocentric |
| 9 | Miami | Monocentric | Super Monocentric |
| 10 | San Francisco | Monocentric | Super Monocentric |
| 11 | Melbourne | Fragmented | Super Monocentric |
| 12 | Sydney | Monocentric | Super Monocentric |
| 13 | Dubai | Fragmented | Super Polycentric |
| 14 | Doha | Fragmented | Super Monocentric |
| 15 | Riyadh | Fragmented | Semi Polycentric |
| 16 | Tel Aviv | Fragmented | Semi Monocentric |
| 17 | São Paulo | Fragmented | Semi Polycentric |
| 18 | Panama City | Fragmented | Semi Polycentric |
| 19 | Mexico City | Fragmented | Semi Polycentric |
| 20 | Nairobi | Fragmented | Semi Monocentric |

The Hudson Yards development in New York (under construction) clusters mixed-use towers around mass transit. This massive 18-million-square-foot complex will create a vibrant socio-economic magnet and form an important focal point in Manhattan's West Side. At a cost of $20 billion, it will contain a cluster of skyscraper masterpieces including: 10 Hudson Yards, 15 Hudson Yards, 30 Hudson Yards, 35 Hudson Yards, 50 Hudson Yards, 55 Hudson Yards, 15 Penn Plaza, 3 Manhattan West, 3 Hudson Boulevard, and Hudson Rise Hotel. In addition to marvelous engineering (required to construct skyscrapers over active large rail yards), the new complex will integrate state-of-the-art technologies (e.g., building management tools, security technology, and elevator systems). Similarly, in Beijing, 20 tall buildings, ranging from 150 m to 350 m in height, will cluster around Zun Tower, which will rise to 528 m (1732 ft) and become the tallest building in Beijing and the eighth tallest building in China.

In the same vein, London has embraced a policy of clustering tall buildings around key rail stations and creating mixed-use "spatial magnets" [30]. London's neighborhoods have been attempting to create social life and vibrancy after office hours and during weekends by integrating residential developments, retails, amenities, services, and transit centers. In this regard, David Nicholson-Cole explains, "To cope with the pressures of dynamic mass-transit systems and rising land values, urban citizens must grow accustomed to living—as well as working—in high-rise developments, clustered around key transport nodes" [48]. Union Square in Hong Kong has clustered distinct high-rises including ICC—the city's tallest building—within a small area of about 30 acres (12 hectares). Internally, the iconic transit station gives Union Square a unique identity. Similarly, in Dubai, planners have clustered the world's tallest building (Burj Khalifa), world's biggest mall, and largest "dancing" fountain to give this section of Dubai's downtown a special character.

## 5. Conclusions

This paper elucidated the massive construction boom of tall buildings globally. All major cities in all the world's continents are actively constructing tall buildings. As such, planners, architects, and politicians must carefully study these developments because they are profoundly reshaping

these cities. Tall buildings are urban planning game changers that alter the spatial patterns and physical character of cities. Planners must engage the public in examining these developments and work together to minimize their negative impacts and maximize their environmental, social, and economic benefits [52].

While it is impossible to provide a definitive scientific proof, this research, based on the provided review, predicts that tall building activities will continue in the near future for multiple reasons. Due to the influx of large numbers of people into cities from rural and suburban areas, numerous cities around the world will need to plan for rapid urban growth and many of them already do. If growth cannot be stopped, then it must be managed. The places that will thrive are those prepared to grow to improve a city's quality of life. The number of megacities with a population of 10 million or more has climbed from 1 in 1950 to 5 in 1975 to 14 in 1995, and to 23 in 2015. In addition, the number is expected to reach 30 by 2020 [39]. Megacities will continue to develop and grow, which is a phenomenon requiring more skyscrapers to avoid urban sprawl and other associated problems [53,54].

In the advent of rapid urbanization, urban theorist Carol Willis explained that cities will have three options: (1) horizontal overcrowding, (2) urban sprawl, and (3) vertical expansion. She views vertical expansion as the prevailing choice [21]. This view is shared by other urban theorists, architects, and urban designers such as Ken Yeang, Roger Trancik, Richard Rogers, and Trevor Boddy [28]. In addition, in this regard, Heller et al. wrote, "The forces which are shaping cities worldwide, related to the effects of population growth and urbanization, are compelling and unavoidable. In the near term, we should continue to see an expanding role for tall buildings in our urban fabric" [55] (p. 379).

Sustainability or sustainable development is increasingly making its way to the top of the agenda of planning and urban design policies and regulations. Sustainability promotes dense and mixed-use living that reduces travel time and carbon emission. Although vertical density is not the only way to achieve dense living, it is a viable option especially in areas where land is in demand and expensive [56,57].

Concerning the near future (10–15 years), numerous tall and supertall buildings are under construction and in the planning stage worldwide. As this review reveals, in China, numerous first-tier cities including Beijing, Shanghai, Shenzhen, and Nanjing are building high-rises and second-tier and third-tier cities are following suit. The same is happening but to a less extent in India and Japan. Furthermore, major U.S. cities such as Chicago, New York City, Miami, and Houston continue to build and plan to build significant high-rises [58].

Regarding the distant future, no one has a crystal ball to see what will happen 50 to 100 years from today, but it can be predicted that urban population will increase as explained earlier. Therefore, we may ask an important question: Can cities expand laterally without sprawling and destroying valuable agricultural land? The migration of rural population to cities in developing countries will continue unabated. People want to move to where the din and bustle of other people, a flurry of activities, social and recreational services, and above all, employment opportunities are. Because of this, once people become comfortable with urban living, they will likely embrace it for a long time.

The tall building and the city are intertwined and their prospects are tied together. When a tall building is constructed, it must work well with the city both at present and in the future. Both tall buildings and cities are multi-faceted and multi-disciplinary. They both have a common goal of realizing a sustainable future for their posterity. Therefore, the integration of their respective systems and professional collaboration among planners, architects, engineers, developers, and contractors as well as cooperation and support of the city officials and the public at large are all essential. For the 21st century, it can be predicted that skyscrapers will have a significant international role in the global village [59,60].

## 6. Future Research

This paper offers the groundwork for many future studies on cities and tall buildings nexus. Each examined city requires deeper analysis and investigation of the various socio-economic, population, and demographic variables. Comparative studies of cities within or across continents

are also needed. Other studies may focus on understanding the role of tall buildings in reshaping spatial patterns of their cities. Importantly, this paper could serve as a template for future research that reviews tall building developments on a regular basis.

**Funding:** This research received no external funding

**Acknowledgments:** The author would like to thank the CTBUH for useful skyscraper database. However, it is important to acknowledge that the database has limitations. For example, it does not include all buildings below 150 m and the generated maps display buildings with GPS recordings only.

**Conflicts of Interest:** The author declares no conflict of interest

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
