# Peer review of "Skyscrapers in the Twenty-First Century City: A Global Snapshot"

_buildings, doi:10.3390/buildings8120175_

Round 1

Reviewer 1 Report

Skyscrapers in the Twenty-First Century City: A Global Snapshot

The paper provides a review article, exploring in depth construction trends prior to 2000, and at 2020 – to explore the changes in tall buildings across these 20 years.

The research heavily relies on the CTBUH’s Skyscraper database, which is fine, but the analysis provided is far too descriptive, lacking cutting analysis to explain why these construction trends have occurred and what has fuelled them.

On line 27 the authors note “for answering these questions, the paper conducts an extensive literature review”. There is actually a very limited literature review, especially for a ‘review paper’. This should be expanded on in section 2. Additional literature could include:

·         Belinda Yuen. High-Rise Living in Asian Cities

·         Oldfield, Trabucco and Wood 2008. Five Energy Generations of Tall Buildings

·         Graham Vertical: the City from Satellites to Bunkers

·         Graham and Hewitt, 2012 Getting off the ground: on the politics of urban verticality

·         Harris, 2015 Vertical urbanisms: opening up geographies of the three-dimensional city

·         Sassen, 2000 Cities in the Global Economy

·         Robert Tavernor Visual and cultural sustainability: The impact of tall buildings on London

Note: these are all peer review papers or published books of high quality – and there are many more available. I would expect an extensive literature review for a review paper.

You note that geographically Asia is the centre of tall building construction, numerically at least (line 157). Why is this? Asia also has the greatest levels of urbanisation, some of the greatest changes in finance, population, and other development metrics. The analysis is far too descriptive, giving us the trends, but not articulating the drivers. There is perhaps a little of this between lines 200 – 215, but I would expect more.  

Are the maps at the same scale? What do you mean smaller / larger geographic scale (line 35). The mapping analysis shows promise, demonstrating not just numerical spread, but geographic and morphological trends too. The maps are startling visually, in showing growth of the typology. However the maps are just pictures, and there is little analysis of them. Such a study could identify clusters of towers, links with transit, links between tower development and changes in regulation – for examples, towers in Pudong have been fuelled by the state defining the Lujiazui area as the only finance and trade zone. London’s tall buildings are spread in clusters because of the LVMF regulation. New York’s have been driven by the opportunities of smaller sites (super slender towers) becoming financially viable, and sites being able to purchase zoning heights from other sites – and hence create supertall towers. There is some categorisation of tall building city forms (mono-centric, polycentric, etc), but little discussion around this.

It might be, to get the level of depth needed in analysis, you need to explore less cities, but in a greater level of detail?

It is worth noting too, the CTBUH database has limitations – the database itself acknowledges that it does not include all buildings below 150m. This needs to be acknowledged in the paper.

Reviewer 2 Report

The paper presents worldwide statistics related to the growth in skyscrapers. Although it is a well-written paper, the reviewer suspects if it is a good fit for a scientific paper. Below are the reasons (some might overlap):

-All the statistics presented in the paper is taken from CTBUH which seems to have the entire contribution. Extracting these statistics from a single database and rearrangement without further work does not have a publication value according to the reviewer. 

-Likewise, there is no novelty, no new finding presented in the study. Such statistics are already available to public as the authors stated, and one can access the database and collect the information without extensive effort.

-There is no significant conclusion, everything drops down to skyscrapers being built at a growing rate. This is common knowledge not only for engineers and scientists, but also for public.

-To add value to the database, one needs to do advanced statistical analysis according to the reviewer which does not exist in the manuscript. 

-Some cases are selected in the study, while others left behind. It is very likely that the choice of paper's scope might be questionable. This would be a typical response from a reader, the reviewer would like to state it in advance.

-Finally, if this is a review paper, the literature review is extremely small according to the reviewer.

Reviewer 3 Report

The paper contains an analysis of the tall buildings that have been built worldwide in the last few years in terms of their geographical location, height and use - offices, residential etc.

The paper does not attempt to analyse the technical difficulties in constructing very tall buildings, nor does it address issues such as sustainability or transport in detail. This is not a criticism, the data it presents is sufficient in itself.

There is a large amount of data, and the fact that some heights are given in metres and feet and some only in metres makes the information more difficult to assimilate. I would suggest using only metres, especially since the 1000m height has special significance. Incidentally technical papers should spell the unit as metre, not meter, since it is an internationally agreed unit.

Reviewer 4 Report

The main body of the paper is composed of summaries of extensive tall building data. As the main body is very long (perhaps too long), the discussion section is in a sense also a summary of the summaries presented in the main body. It is recommended to include more critical analyses of the summaries.

Reviewer 5 Report

Dear Authors, 

   Thank you for your efforts in preparing this manuscript. I am pleased to review your work and would like to share my suggestions for improving the final draft. 

#1: There are many tables, but a few figures in your paper. You may change some of the tables into figures. For famous towers, you can find the sketches to add to your plots (example: https://vincentloy.wordpress.com/2009/01/13/top-5-tallest-telecommunication-towers-in-the-world/). 

#2: The maps need to be provided in vector format or with higher resolution. Also, all the figures need to have a-b-c-... labels on them. 

#3: Change the lists into x-y plots for a better presentation. 

#4: The author's contribution is not evident from the introduction-to-conclusion. Please clearly state that what gaps may this study fill or solve. Otherwise, all the maps, tables, and the figures can be found on the Internet. Please clearly state the novelty of your research in the introduction section.

#5: The references should be improved with the relevant recent studies. A list of websites is not recommended. 

Regards,

Round 2

Reviewer 1 Report

The new section of ‘driving forces’ adds significantly to the quality of the paper. The research includes a stronger literature review, which is beneficial also. You mention key cities that are building ‘sustainable’ tall buildings due to the need to make cities more compact (bottom of page 8, beginning of page 9). However, I question some of these. Is the Gold Coast really trying to use towers to create compact living? Sydney is far less dense than say, London. In these cities tall buildings are driven by land costs more than anything else. The discussion on the 2009 Kyoto Protocol has been superseded by the recent Paris agreement (see section 2.8). It would be useful to include some evidence on compactness, verticality and lower carbon emissions here too (for example, see the work of Peng Du and Antony Wood www.mdpi.com/2075-5309/5/3/1003/pdf)

Reviewer 2 Report

The authors' modifications are appreciated however, I still doubt if the material is suitable as a research paper. The data analytics addressing this scope should be deeper I think, considering the vast amount of statistics available online.

Other than that, one problem I observe is that too many data tables without a particular point and merit is overloading the paper. 

On the other hand, it has a considerably good shape as a review paper after the changes made, yet, I am skeptical if a review addressing tall building height is essential at the current stage. 

Reviewer 5 Report

Dear Authors, 

   Thank you very much for the revisions and for the additional sections that you have added to your original paper. After considering the other reviewer's comments, as well as reviewing for possible writing errors,  I would recommend the paper for publication. Please pay attention to the page numbers limitations in your final draft, as well. 

Regards,